# Are you with me? Co-occurrence tests from community ecology can identify positive and negative epistasis between inversions in *Mimulus guttatus*

**Luis J. Madrigal-Roca**[ORCID]*, **John K. Kelly**[ORCID]

Ecology and Evolutionary Biology's Department of the University of Kansas, Lawrence, Kansas, United States of America

* madrigalrocalj@yahoo.com

## Abstract

Chromosomal inversions are structural genetic variants that can play a crucial role in adaptive evolution and speciation. Patterns of attraction and repulsion among unlinked inversions — whether they tend to be carried by the same or different individuals— can indicate how selection is acting on these polymorphisms. In this study, we compare analytical techniques using data from 64 inversions that segregate among 1373 $F_2$ plants of the yellow monkeyflower *Mimulus guttatus*. Mendelian assortment provides a strong null hypothesis for $\chi^2$ contingency tests. Here, we show how co-occurrence metrics used in community ecology can provide additional insight regarding coupling and repulsion of inversions at genotypic level. The centered Jaccard/Tanimoto index and the affinity score describe the specific way that inversions interact to generate epistasis for plant survival. We further explore the use of network analysis to visualize inversion interactions and to identify essential third and fourth order interactions, which expand the traditional pairwise scope of the co-occurrence metrics. We suggest that a combination of different statistical approaches will provide the most complete characterization of the fitness effects, both for inversions and other polymorphisms essential to adaptation and speciation.

## Introduction

Co-occurrence analysis traces its origins to the 18th century, evolving from probabilistic frameworks to the development of contingency tables and formal statistics in the 20th century [1]. This analytical approach now serves as a fundamental tool across biological disciplines, quantifying similarities between diverse sample types including community observations, diseases, and genes. In both ecology and genetics, co-occurrence analysis has become instrumental in decoding complex interaction patterns, building on the hypothesis that co-occurring entities share functional relationships [2]. The approach illuminates community and landscape-level interactions in ecology, while in genetics, linkage disequilibria statistics characterize non-random relationships between *loci* as examples of co-occurrence metrics [3]. Though terminology varies across disciplines, these associations share theoretical foundations, enabling the reciprocal application of ecological statistics to genetic data and vice versa.

**Data availability statement:** Data included in public repository: https://github.com/luismadrigal98/Inv_cooccurrence_Mguttatus

**Funding:** This research is funded by one NSF grant received by Dr. John K. Kelly, which is MCB-1940785. The funders had no role in study design, data collection and analysis, decision to publish, or preparation of the manuscript.

**Competing interests:** The authors have declared that no competing interests exist.

The mechanisms driving co-occurrence differ between community ecology and genetics. Species coexistence or repulsion across locations may stem from environmental preferences [4], biotic interactions [5], or historical artifacts [6]. In the genetic context, associations between alleles at different *loci* can arise from physical linkage or through natural selection maintaining associations when *loci* interact to determine fitness. This paper explores statistical approaches for detecting and characterizing associations, leveraging an experimental design where co-occurrence among genetically unlinked *loci* must result from differential survival or gametic selection. This methodology offers fresh insights into genomic data relationships, paralleling applications in ecological studies and advancing our understanding of genetic interactions and regulatory networks.

In genetics, co-occurrence is usually characterized as linkage disequilibrium—the non-random association between alleles at two genetic *loci* [3,7]. The original measure, D, quantifies the covariance of allelic state (0 or 1) between *loci* across gametes for biallelic *loci*. This haploid-level metric [8] spawned normalized derivatives such as D' [9] and $r^2$ [10]. These statistics score associations at the allelic level, which is particularly significant given that genotype data typically comes from diploid individuals where the phase of alleles at different *loci* remains unknown. We examine the centered Jaccard/Tanimoto index (cJ/T) [11,12] and the affinity score [13] as measures of association at the diploid genotype level. The cJ/T evaluates associations by comparing observed co-occurrence against independence-based null expectations, revealing positive (cJ/T > 0) or negative (cJ/T < 0) associations. The affinity score, expressed as a log odds ratio (α), similarly captures co-occurrence patterns, with positive values indicating coupling and negative values suggesting genetic repulsion. These metrics establish a sophisticated framework for analyzing complex genetic interactions, particularly valuable where traditional LD measures prove inadequate.

The co-occurrence metrics we examine are applicable to any genetic polymorphism, though we focus on chromosomal inversion genotype data. Inversions represent large-scale structural variants that can encompass substantial chromosome portions, containing more genetic material than single nucleotide polymorphisms (SNPs). The Australasian snapper *Chrysophrys auratus* exemplifies this, with inversions encompassing three times more base pairs than reported SNPs [14]. As adaptive characters subject to selection, inversions frequently exert significant fitness effects [15]. In our study species *Mimulus guttatus*, a 4.5mb inversion on chromosome 6 influences both morphological traits and fitness measures [16]. While inversions are known to suppress recombination between closely linked, co-adapted *loci* contributing to adaptation [17,18] less research has addressed how unlinked inversions segregating within the same population might affect fitness. Our recent work identified 64 inversion polymorphisms segregating within a large experiment using crosses between inbred yellow monkeyflower lines [19]. We now evaluate interaction effects (epistasis in genetic terminology), specifically examining whether inversions generate genetic incompatibilities—negative interactions that reduce fitness or cause lethality when different inversions co-occur [20].

Pairwise association tests face an inherent limitation in examining only two entities simultaneously, despite biological systems often involving complex, multi-way interactions. Network analysis has emerged as an increasingly important tool in biology [21,22], offering an intuitive representation of complex data that emphasizes entity interactions. This association mapping approach represents entities (*e.g.,* genes and structural variants) as vertices or nodes, with their interactions depicted as links or edges [23]. By synthesizing pairwise tests, network analysis provides a comprehensive view of the interaction landscape. In this paper, we demonstrate how network analysis can reveal higher-order interactions among genetic *loci* in determining plant survival. The combination of co-occurrence indexes and network analysis

enables us to transcend pairwise comparisons and understand the complex interplay between multiple inversions.

Using the *Mimulus* inversion dataset, we investigate whether combining ecological co-occurrence analysis with network analysis can reveal novel patterns of positive and negative epistasis —coupling and repulsion in linkage nomenclature. We focus on inversions on different chromosomes, ensuring that any detected associations must arise from differential plant survival. We also examine individual inversion effects on survival through segregation distortion analysis and explore potential "domino effects" on subsequent co-occurrence analysis. These patterns illuminate the functional relationships and coevolutionary dynamics of inversions in *Mimulus*. Understanding inversion interactions advances our knowledge of adaptation mechanisms, speciation, and genome evolution in plants. Furthermore, our methodological approach, which integrates ecological and genetic perspectives with network analysis, establishes a valuable framework for studying complex genomic interactions across species, potentially advancing our understanding of adaptation and evolution across diverse *taxa*.

## Materials and methods

### Data preparation and inversion identification

The raw data analyzed in our study were published by [24] as part of a replicated F2 mapping experiment using inbred lines of the yellow monkeyflower, *Mimulus guttatus*, from the Iron Mountain (IM) population in Oregon's Cascade Mountains (44.402217N, 122.153317W). From a panel of 187 whole-genome sequenced inbred lines, they selected a reference line (IM767) and nine alternative lines that were genetically unrelated to both IM767 and each other (based on genome-wide nucleotide diversity). Each alternative line was crossed with IM767 (as pollen donor) to generate F1 hybrids. A single F1 plant from each cross was self-fertilized to produce F2 seeds. Both parental lines and F2 progeny were grown in controlled greenhouse conditions. All parental lines were scored for orientation at 64 inversion polymorphisms [19]. For each segregating inversion in a cross, F2 individuals were classified as RR (homozygous for the reference orientation), RI (heterozygous), or II (homozygous for the inverted orientation). We present the complete genotype matrix for F2 individuals from the nine crosses as Supplemental Excel Spreadsheet 1 for future reference. The number of F2s varied among the crosses: IM62 (n = 92), IM155 (n = 188), IM444 (n = 148), IM502 (n = 163), IM541 (n = 164), IM664 (n = 157), IM909 (n = 148), IM1034 (n = 151), and IM1192 (n = 162). Since not all inversions segregate in all crosses, we detected 941 unique valid combinations from the 2016 possible pairs of inversions (64 choose 2), representing 47% of the combinatorial space. When each inversion was decomposed into heterozygous or homozygous states and their presence/absence analyzed in a genotypic context, the combinatorial space expanded to 8128 possible combinations. Of these, we detected 3764 valid contrasts (approximately 46% of the combinatorial space) between *loci* on different chromosomes. All analyses were performed using R 4.4.2 [25].

### Segregation distortion analysis for individual inversions

To test for segregation distortion (SD) at individual inversion *loci* (Fig 1A), we performed a goodness-of-fit test to determine if inversion genotype frequencies (RR, RI, II) followed Mendelian proportions for an F2 population (0.25/0.50/0.25). We used the GTest function from the R package DescTools [26], applying the Williams' correction. We also characterized the pattern of selection in cases of SD by employing a sequential analysis proposed by [27]. We used the segregation distortion value (SDV), defined as the negative decimal logarithm of the SD-associated *p*-values, to explore the relationship between individual tests and the pairwise

**A)  Goodness-of-fit test for segregation distortion (SD) based on G statistic**

**Genotypes**

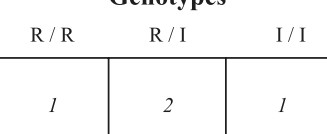

Mendelian expectations (ratios)

$$G = 2 \sum_i O_i \cdot \ln\left(\frac{O_i}{E_i}\right)$$

Degrees of freedom: 2

**B)  $\chi^2$ testing framework (3 x 3 table)**

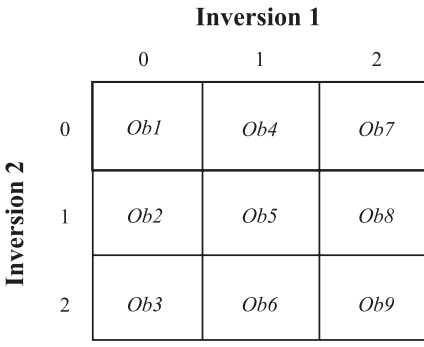

$$\chi^2 = \sum_{c=1}^{9} \frac{(Ob_c - Ex_c)^2}{Ex_c}; \text{ where } Ex_c = f * N$$

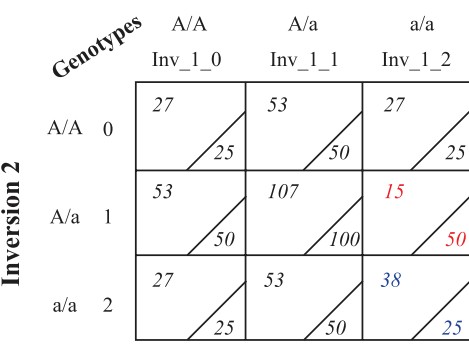

A: Standard orientation a: Inversion

Inflation in counts Deflation in counts

Degrees of freedom: 4 N (specific per line)

**C)  Co-occurrence framework (2 x 2 tables)**

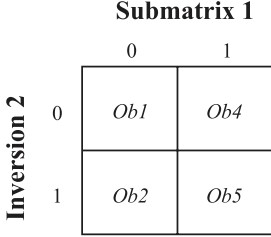 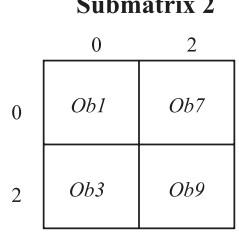 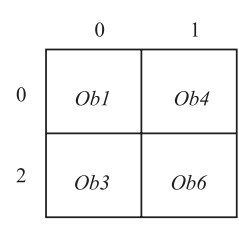 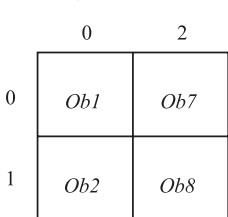

**Inversion 1**

Centered Jaccard / Tanimoto Index Degrees of freedom: 1 N (specific per line)
Affinity score

**Fig 1.  Relationship between the tabulated information employed for hypothesis testing under the different approaches presented in this study. A**. Goodness-of-fit approach to assess the presence of segregation distortion (SD) for each inversion. **B**. 3x3 table framework used for performing $\chi^2$ tests of independence between dosage levels of pairs of inversions. Note that *f* referrers to the frequency of each genotype following Mendelian proportions, and *N* is the total number of individuals. To the right is depicted an example scenario where a deviation from independence exists. In that example, below the diagonal there are tabulated the expected counts and above it the observed ones. With red color is represented a case of deflation of the counts in relation to the expectation (repulsion) and in blue a case of inflation in the counts (coupling). **C**. 2x2 table framework used as basis for co-occurrence analysis with ecological indexes.

contrasts performed subsequently. This association was evaluated because segregation distortion involves complex relationships between genes, which can shape the transmission advantage of certain haplotypes (determined in our context by the combination of inversions) and affect their genetic associations.

## Contingency table analysis

We applied contingency analysis to each pair of inversions segregating within each cross to evaluate the null hypothesis of independence between inversion genotypes. The independence between each pair of inversions was evaluated using an omnibus $\chi^2$ test (applied to the 3×3 table in Fig 1B, left). For reference, a representation of the data structure is provided in Fig 1B for 400 individuals (right). We used chisq.test from the base R package and computed the *p*-value associated with the $\chi^2$ statistic through bootstrapping (10,000 iterations). We used bootstrapping because small expected frequencies were observed in some tests which can cause poor approximation of the $\chi^2$ statistic to its theoretical distribution. Additionally, we employed the standardized residual method and analyzed the relative contribution of each cell in the 3×3 table to assess between which dosages of inversions there were significant deviations from expectations. For this analysis, we used the chisq.posthoc.test function from the R package of the same name [28].

## Co-occurrence metrics

Two different indexes were employed for quantifying the co-occurrence and repulsion patterns: the centered Jaccard/Tanimoto index (cJ/T) [11] and the affinity score described recently by [1]. The cJ/T index and the hypothesis testing framework for it were analyzed using jaccard.test.pairwise from the jaccard R package [12]. The *p*-values were estimated using the bootstrap method (B = 10,000 replicates), which provides an accurate and efficient framework of analysis when compared with other available approaches [11]. For the affinity estimation, we used the Co-occurrenceAffinity R package [13], specifying the data type as binary in the Co-occurrenceAffnity function. In the context of this study, these two indices decompose the 3×3 contingency table and perform hypothesis testing on the information contained in the four submatrices reported in Fig 1C. Given the lower number of degrees of freedom of the 2×2 tables, the tests based on these should be less powerful in terms of signal detection but inform the direction of co-occurrence patterns.

We utilized these two co-occurrence frameworks because both are 0-centered, which naturally facilitates the identification of coupling or repulsion patterns between the inversions analyzed (Fig 2). The affinity score method provides an important statistical advantage over conventional similarity indexes (uncentered Jaccard/Tanimoto, Sørensen-Dice, and Simpson). When using traditional indexes, the strength and direction of associations can be distorted by how frequently entities occur in the dataset. Affinity score eliminates this prevalence bias [1], producing consistent measurements of association strength regardless of whether the analyzed entities are common or rare in the population. Nevertheless, it is important to note that these measures are also susceptible to false discoveries common in genomic applications, and careful interpretation is required in cases where independence between tests is not achieved (*e.g.*, there will be four tests associated with the same pair of inversions, as seen in Fig 1C after decomposing the 3×3 table depicted in Fig 1D).

Furthermore, a significant distinction from traditional linkage disequilibrium (LD) measures is that these two indexes enable us to analyze the relationship between inversions at the genotypic level (Fig 2). In contrast, traditional methods, such as D and D', focus on a broader allele scope through haplotypes. The genotypic scope is preferred when using Chi-squared

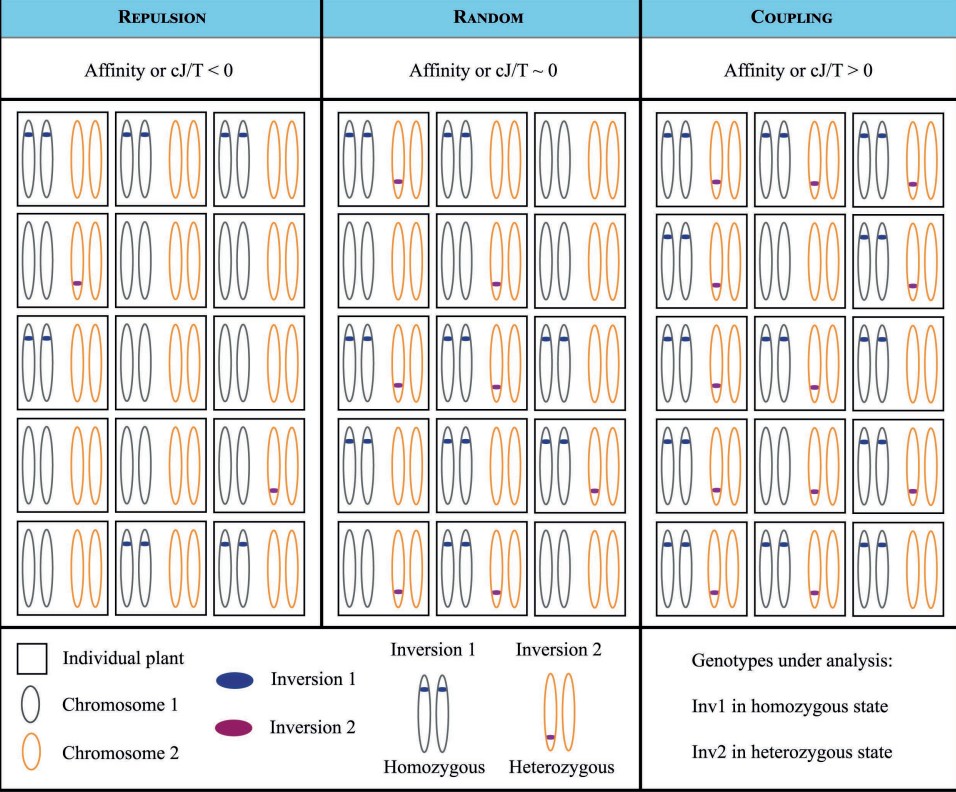

**Fig 2. Patterns of inversion genotypes co-occurrence across F2 individuals of *Mimulus guttatus* explored through the affinity and centered Jaccard/Tanimoto indexes.** Each panel shows a representative hypothetical pair of chromosomes from diploid individual plants (squares), illustrating three scenarios of association between Inversion 1 (in homozygous state) and Inversion 2 (in heterozygous state). In repulsion (left panel), inversion genotypes occur together less frequently than expected by chance (Affinity or centered Jaccard/Tanimoto < 0), suggesting potential negative fitness effects of their co-occurrence. Under random association (middle panel), inversions assort independently (Affinity or centered Jaccard/Tanimoto ≈ 0), following Mendelian expectations. In coupling (right panel), inversion genotypes co-occur more frequently than expected by chance (Affinity or centered Jaccard/Tanimoto > 0), indicating possible positive epistatic interactions (coupling). Blue and purple segments represent inverted regions. Gray and yellow ellipses represent the chromosomes, and two per individual are depicted according to the ploidy level of our model plant.

approaches and odds ratios tests to avoid potential biases introduced by summing allele frequencies [29]. Nevertheless, the main reason we avoided direct use of LD metrics relates to their variability with the allele frequencies at each *locus* and linear to non-linear shifts in the LD structure between *loci* [8]. We included a simulation-based contrast between two widely used LD metrics (D and D') and the co-occurrence indexes employed in this study, which is explained in the Extended Methods and presented as Supporting Information (S3 Table in S1 Word Doc and S4 Fig).

## Stochastic simulations demonstrate the ability of alternative approaches to detect interactions

To assess the behavior of different statistical frameworks (one global approach using 3×3 tables and two specific approaches using 2×2 tables), we developed a simulation in R 4.3.2 [30] using the script Aux1_Contingency_tables_simulation. The null model distributed 400 individuals across 9 cells of a contingency table, representing combinations of genotypes from

two inversions (Fig 1B, left). We populated 100,000 simulated observed tables using Mendelian expectations ([Fig 1B](), right). These simulated tables were compared against expected counts using an omnibus $\chi^2$ test. Each table was also decomposed into four submatrices, with information transformed into binary vectors (presence/absence) to apply Jaccard and affinity measures. For each generated table, we obtained and stored one global *p*-value from the omnibus $\chi^2$ test and four *p*-values per co-occurrence index (one per submatrix). We then analyzed the distribution of all these *p*-values.

We simulated two alternative scenarios beyond the null model. Expected frequencies were modified by multiplying the Mendelian expectation by a survival rate and normalizing by dividing by the sum of that product in each cell (Equation 1). The survival rate represents the probability that an individual with a given genotype survives to be observed relative to Mendelian inheritance expectations. A survival rate of 1 indicates no deviation from Mendelian expectations, while values below 1 indicate reduced survival (deflation). In the first alternative model, we deflated the 9th cell counts using a survival rate of 0.2 (relative survival for other cells was 1). In the second model, we deflated the 8th cell counts (survival rate 0.2), inflated the 9th cell counts (survival rate 1), and set the relative survival for remaining cells at 0.7. Statistical tests were performed as previously described, retaining only the *p*-values for distribution analysis. We visualized frequency histograms and cumulative distribution functions using the ggplot2 framework in R [31].

$$f_{e;mod} = \frac{f_e * survival\ rate}{\sum_{i=1}^{9}\left(f_e * survival\ rate\right)}$$

(1)

Where $f_e$ is the Mendelian expected frequency, and $f_{e;mod}$ is the expected frequency modified by the survival rate. These specific survival rates were chosen to test the ability of each statistical method to detect deviations from additivity. This allows us to establish proof-of-principle for our methods before applying them to real data, where the deviations may be more subtle.

## Congruence between methods

We explored the level of congruence among statistical frameworks by linking values from the standardized residuals derived from the $\chi^2$ *post hoc* test, the cJ/T, and affinity scores through adjusted linear models using the R base function lm. Since the residuals did not follow a normal distribution in any of the paired models, we implemented robust linear regression using the rlm function from the MASS package [32]. We estimated the *p*-values associated with model coefficients using the rob.pvals function from the repmod R package [33]. After visualization, we removed outliers using a cutoff threshold of 0.15 for the weights derived from the regression model.

## Network visualization and analysis

The pairwise centered Jaccard/Tanimoto indexes served as reference for building interaction networks using the igraph [34] package in R. We used only one of the co-occurrence indexes since they are linearly related (see results below). Networks provide an effective visualization of repulsion and coupling patterns, alongside other metadata such as non-significant relations volume, chromosomal membership of the variants studied, etc. We considered patterns significant when raw *p*-values of the centered Jaccard/Tanimoto (used for weighing network edges) were smaller than 0.05. Edges belonging to significant omnibus $\chi^2$ tests were highlighted to demonstrate the difference between significant and non-significant contrast weights according to the establish convention for this study.

We then extracted subgraphs defined by edges with raw index-associated *p*-values less than 0.05. Using these subgraphs, we assessed for significant third and fourth order motifs (e.g., triangles, cycles, etc.). In this context, a motif represents an association pattern defined by nodes connected in a specific manner with frequency higher than expected by chance [35,36]. Node significance was assessed through a permutational procedure that rewired each network multiple times, counting motif appearances. This generated a null distribution of counts per motif, allowing us to determine the probability of observing counts as extreme or more extreme than those observed under the null hypothesis (randomness). For simplicity, we evaluated only overrepresented motifs with a minimum count of 200, excluding scarce motifs. With null distributions established, we determined the most significant motifs across networks (as the negative decimal logarithm of adjusted *p*-values), the observed-to-expected ratio, and Z-scores associated with observed counts.

### Heterogeneity test for the shared inversions (study case)

Three inversions (29, 32, and 40) segregate in all nine crosses. For these inversions, we generated networks to visually inspect whether configurations were variable or conserved. We employed a replicated G-based goodness of fit test [37] to assess whether relationship patterns were conserved or variable across the nine genetic families (crosses). For both 3×3 and 2×2 table frameworks, we determined the $G_{total}$, $G_{pooled}$, and $G_{het}$ statistics and contrast them against the $\chi^2$ distribution with appropriate degrees of freedom.

### Relationship between SD and co-occurrence patterns

We also assessed if inversions showing significant segregation distortion were more or less likely to be involved in interactions with other inversions. For this, we used the results from the omnibus $\chi^2$ independence tests and the SDV. We explored the Spearman correlation between an inversion's SDV value and two metrics: (1) the frequency with which the inversion had raw *p* values below 0.05 in independence tests with other inversions, and (2) the mean *p* value of such tests. Additionally, we fitted a generalized linear model to analyze the number of significant tests as a function of the SDV, controlling for genetic family (cross), using a quasi-Poisson family in R's GLM framework.

### *p*-values correction

We implemented multiple methods to control false discoveries in our statistical analyses. For the omnibus $\chi^2$ test, we applied the Benjamini-Hochberg procedure to correct *p*-values within each line [38]. We also employed this method for FDR control during the motif discovery phase of our network analysis. For *post hoc* tests, we implemented the Benjamini-Yekutieli method since the contrasts from the 3×3 table are not independent [39].

However, these corrections proved problematic for our pairwise co-occurrence metrics and the omnibus $\chi^2$ test. Standard methods were excessively conservative given our study design: with 150–200 F2 individuals per cross and our bootstrap-based omnibus test, there exists an inherent lower limit to possible *p*-values for each cross and inversion pair combination. Traditional family-wise error rate controls (like the Bonferroni correction) were also overly conservative given the substantial number of interdependent pairwise tests. We attempted to use the q-value approach described by [40] to control the FDR. However, this method relies on p0 (the estimated proportion of true null hypotheses), which requires a uniform distribution of *p*-values above 0.5. Our *p*-value distribution violated this assumption, rendering the method unsuitable for our analysis.

# Results

## Testing of methods via simulation

Regarding the co-occurrence tests, simulations confirmed the close relationship between our different analytical approaches. Under the null model of independent assortment (Mendelian expectations), the omnibus $\chi^2$ test and all co-occurrence indexes (centered Jaccard/Tanimoto index and affinity score) yielded results consistent with expected *p*-value distributions. In all cases, we retrieved the uniform distribution, although slight deviations were detected due to the discrete nature of the data. Nevertheless, analysis of the cumulative distribution function and frequency histograms demonstrated the appropriate behavior of all statistical frameworks (Fig S1).

Under the first alternative scenario, where double homozygotes for the inversions had lower viability, the *p*-values derived from the omnibus $\chi^2$ test showed an evident deviation from the uniform distribution (Fig S2A), with a shift toward more extreme and lower *p*-values compared to the previously described null distribution. Similarly, the centered Jaccard/Tanimoto index and affinity score for submatrix 2 (containing the deflated cell counts) also showed a shift toward lower *p*-values (Fig S2C, G), while preserving the null distribution in the other submatrices (Fig S2B, D, E, F, H, I). For the final simulation, an equivalent pattern was observed, with the histograms and cumulative distribution plots showing that two submatrices (sub2 and sub4) exhibited deviation from the null expectations for the centered Jaccard/Tanimoto index and affinity score, effectively detecting both introduced perturbations (Fig 3C, E, G, I).

## Segregation distortion of inversions

From the 64 inversions analyzed in this study, 10 showed a significant pattern of segregation distortion in at least one cross (Table 1). The most extreme case is inversion 49 in cross 444, where homozygotes for the inversion were completely absent among F2 plants. This pattern suggests that the allele is a recessive lethal, but remarkably, this same inversion occurs frequently in homozygotes in six other crosses and it is homozygous in the IM444 line itself, which is a viable and fertile genotype. Inversion 53 is significantly distorted in three crosses, although the pattern of distortion varies (inverted homozygote deficiency in two cases, reference homozygote deficiency in the other). All other cases in Table 1 are inversions that are distorted in only one cross. Finally, all significant cases favor the zygotic selection model over the gametic model, although this may reflect restrictive assumptions for sequential test employed to detect them.

## Co-occurrence of *Mimulus* inversions

Applying the tests to the *Mimulus'* data, we find that 73 of 941 valid contrasts (~ 8%) yielded raw $p < 0.05$ for the omnibus $\chi^2$ test. Among contrasts that distinguish between homozygous and heterozygous inversion states, we detected 355 significant associations (~9% of valid combinations) using at least one of three analytical approaches: cJ/T index, affinity coefficients, or $\chi^2$ *post hoc* tests (based on uncorrected *p*-values). Nevertheless, a lower number of contrasts were detected as significant (Table 2) for all 2x2 table-based approaches and the omnibus $\chi^2$ test (67, accounting for ~ 2% of the valid combinatorial space). S2 Table in S2 Word Doc shows all the valid contrasts that were detected by at least one of the 2x2 table-based approaches employed here. Importantly, there are 174 cases where the omnibus $\chi^2$ test fails to detect a signal, but the 2x2-based analyses capture them.

As a general summary, aggregating results that strongly deviated from the null hypothesis and were detected by all three 2×2-based approaches (67 in total), we observed distinct

**Table 1. Significant segregation distortion for inversions segregating in a natural population of *Mimulus guttatus* from Iron Mountain, Oregon, USA.**

| Inv | Cross | RR | RI | II | G | p_value | SDV | $\chi^2_1$ | $p_1$ | $\chi^2_2$ | $p_2$ |
|---|---|---|---|---|---|---|---|---|---|---|---|
| Inv_49 | 444 | 92 | 56 | 0 | 135.77 | $3.29 \times 10^{-30}$ | 29.48 | 114.38 | $1.08 \times 10^{-26}$ | 1518.86 | $0.00 \times 10^{00}$ |
| Inv_55 | 1034 | 53 | 95 | 3 | 64.14 | $1.18 \times 10^{-14}$ | 13.93 | 33.11 | $8.70 \times 10^{-09}$ | 149.71 | $2.00 \times 10^{-34}$ |
| Inv_53 | 909 | 6 | 94 | 48 | 47.92 | $3.94 \times 10^{-11}$ | 10.4 | 23.84 | $1.05 \times 10^{-06}$ | 103.85 | $2.19 \times 10^{-24}$ |
| Inv_53 | 155 | 56 | 117 | 15 | 36.45 | $1.22 \times 10^{-08}$ | 7.91 | 17.88 | $2.35 \times 10^{-05}$ | 77.26 | $1.50 \times 10^{-18}$ |
| Inv_29 | 664 | 33 | 64 | 60 | 13.28 | $1.30 \times 10^{-03}$ | 2.88 | 9.29 | $2.31 \times 10^{-03}$ | 50.74 | $1.05 \times 10^{-12}$ |
| Inv_62 | 909 | 31 | 60 | 57 | 13.07 | $1.45 \times 10^{-03}$ | 2.84 | 9.14 | $2.51 \times 10^{-03}$ | 50.22 | $1.38 \times 10^{-12}$ |
| Inv_43 | 541 | 59 | 78 | 27 | 12.54 | $1.89 \times 10^{-03}$ | 2.72 | 12.49 | $4.10 \times 10^{-04}$ | 56.64 | $5.24 \times 10^{-14}$ |
| Inv_30 | 664 | 28 | 71 | 58 | 12.07 | $2.39 \times 10^{-03}$ | 2.62 | 11.46 | $7.09 \times 10^{-04}$ | 54.71 | $1.40 \times 10^{-13}$ |
| Inv_34 | 444 | 23 | 73 | 52 | 11.49 | $3.20 \times 10^{-03}$ | 2.49 | 11.36 | $7.48 \times 10^{-04}$ | 49.93 | $1.60 \times 10^{-12}$ |
| Inv_53 | 1034 | 41 | 89 | 21 | 11.37 | $3.39 \times 10^{-03}$ | 2.47 | 5.30 | $2.13 \times 10^{-02}$ | 24.06 | $9.33 \times 10^{-07}$ |
| Inv_4 | 155 | 32 | 92 | 64 | 10.92 | $4.25 \times 10^{-03}$ | 2.37 | 10.89 | $9.65 \times 10^{-04}$ | 47.21 | $6.38 \times 10^{-12}$ |
| Inv_25 | 1034 | 28 | 95 | 28 | 10.14 | $6.27 \times 10^{-03}$ | 2.2 | 0.00 | $1.00 \times 10^{00}$ | 10.07 | $1.50 \times 10^{-03}$ |

These inversions are inconsistent with the Mendelian model of segregation (raw *p* values presented for the contrasts were FDR < 0.05). **Inv**: Inversion ID. RR, RI, and II refer to the count per genotype across the crosses. **R**: reference (standard orientation of 767 genetic family). **I**: alternative (inversion). **G**: G-statistic derived from the goodness of fit test with 2 degrees of freedom. **df**: degrees of freedom. **SDV**: Segregation distortion value ( $-log_{10} \, p_{value}$ ). $\chi^2_1$ and $\chi^2_2$ refer to the statistics associated with the sequential tests employed to infer the selection patterns. $\chi^2_1$ describes the balance between the frequency of the two segregating orientations while $\chi^2_2$ describe the distribution of genotypic frequencies. The *p* values associated with them are also reported ( $p_1$ and $p_2$ ). In all cases, the result suggests a zygotic pattern of selection.

patterns in the frequency of inversion combinations. The most frequent deviations from independence occurred in heterozygous-homozygous combinations (~58%), with 20 negative and 19 positive associations (combining both arrangements of homozygous and heterozygous inversions). Double heterozygous combinations showed 16 deviations (9 negative vs. 7 positive, 24%). The least frequent deviations appeared in double homozygous combinations with 12 cases (7 positive vs. 5 negative, 18%).

While most inversion pairs showed no evidence of interaction, several notable exceptions emerged. For example, inversion 35 on chromosome 8 exhibited an interaction with inversion 62 on chromosome 14 within the 62 cross (Table 3, top). This interaction was driven by a strong deficiency of "mixed genotypes" —combinations of heterozygous at one inversion but homozygous at the other— indicated by negative cJ/T and alpha statistics for these genotype combinations. Inversion 35 demonstrated an even stronger interaction with another locus (inversion 43) in the 444 cross (Table 3, lower). However, the nature of this interaction differed, being driven entirely by a lower-than-expected count for the double inversion homozygote.

## Congruence between statistical frameworks

There was strong agreement between the 2×2 table-based methods (χ² *post hoc* approach, cJ/T, and Affinity), as depicted in Fig S3. The most significant signals were simultaneously detected by at least two of the three methods based on the submatrices depicted in Fig 1 (Jaccard/Tanimoto, affinity, and χ² *post hoc* test). This congruence was expected given that robust regression analysis revealed significant linear associations between the three measures, supporting the hypothesis testing under each approach (Fig S3).

**Table 2. Significant contrasts according to the raw *p*-values that were detected by the three 2x2 table-based approaches employed to assess the co-occurrence patterns of pairs of inversions in lines of *Mimulus guttatus* from Iron Mountain, Oregon.**

| INV_1 | INV_2 | $\chi^2$ _global | $p\_\chi^2$ _g | $p\_\chi^2$ _ph | $\chi^2$ _RC | $\chi^2$ _SR | α | α _p | cJ/T | cJ/T_p |
|---|---|---|---|---|---|---|---|---|---|---|
| Cross 1034 | | | | | | | | | | |
| Inv_10a_1 | Inv_29_2 | 10.69 | 0.029 | 0.029 | 16.94 | 2.18 | 0.90 | 0.045 | 0.07 | 0.023 |
| Inv_17_2 | Inv_55_1 | 9.75 | 0.042 | 0.034 | 13.42 | -2.12 | -0.84 | 0.041 | -0.05 | 0.044 |
| Inv_29_2 | Inv_9_1 | 10.69 | 0.029 | 0.029 | 16.94 | 2.18 | 0.90 | 0.045 | 0.07 | 0.022 |
| Inv_35_1 | Inv_58_1 | 10.19 | 0.035 | 0.005 | 19.33 | -2.78 | -0.92 | 0.009 | -0.09 | 0.009 |
| Inv_35_1 | Inv_58_2 | 10.19 | 0.035 | 0.004 | 26.78 | 2.87 | 1.09 | 0.006 | 0.10 | 0.004 |
| Inv_38_2 | Inv_49_1 | 14.31 | 0.006 | 0.002 | 22.08 | -3.16 | -1.17 | 0.002 | -0.09 | 0.004 |
| Inv_38_2 | Inv_49_2 | 14.31 | 0.006 | 0.001 | 44.28 | 3.26 | 1.39 | 0.003 | 0.14 | 0.001 |
| Inv_39_1 | Inv_7_1 | 13.03 | 0.012 | 0.035 | 8.89 | -2.11 | -0.69 | 0.048 | -0.07 | 0.047 |
| Inv_39_1 | Inv_7_1 | 13.03 | 0.012 | 0.035 | 8.89 | -2.11 | -0.69 | 0.048 | -0.07 | 0.047 |
| Cross 1192 | | | | | | | | | | |
| Inv_17_2 | Inv_32_2 | 10.51 | 0.034 | 0.003 | 52.72 | 3.00 | 1.21 | 0.005 | 0.13 | 0.002 |
| Inv_19_1 | Inv_25_2 | 14.05 | 0.007 | 0.005 | 22.87 | -2.84 | -1.09 | 0.006 | -0.09 | 0.010 |
| Inv_23_1 | Inv_61_2 | 9.60 | 0.048 | 0.007 | 27.08 | 2.68 | 1.00 | 0.012 | 0.09 | 0.005 |
| Inv_28_2 | Inv_54_1 | 11.00 | 0.026 | 0.012 | 24.49 | 2.51 | 0.98 | 0.019 | 0.09 | 0.009 |
| Cross 155 | | | | | | | | | | |
| Inv_32_2 | Inv_53_2 | 20.34 | 0.000 | 0.033 | 16.08 | -2.13 | -10.00 | 0.026 | -0.06 | 0.046 |
| Inv_33_2 | Inv_53_2 | 20.36 | 0.001 | 0.026 | 17.24 | -2.23 | -10.00 | 0.024 | -0.06 | 0.036 |
| Inv_35_2 | Inv_58_1 | 10.74 | 0.029 | 0.008 | 23.46 | 2.64 | 1.07 | 0.009 | 0.07 | 0.005 |
| Inv_49_1 | Inv_53_1 | 9.71 | 0.047 | 0.022 | 10.78 | 2.29 | 0.70 | 0.025 | 0.07 | 0.016 |
| Inv_51_1 | Inv_53_1 | 12.54 | 0.011 | 0.003 | 14.30 | 2.97 | 0.92 | 0.004 | 0.10 | 0.003 |
| Inv_53_2 | Inv_58_1 | 12.97 | 0.012 | 0.004 | 26.07 | -2.87 | -1.72 | 0.006 | -0.05 | 0.007 |
| Cross 444 | | | | | | | | | | |
| Inv_16_2 | Inv_49_1 | 9.09 | 0.012 | 0.004 | 43.58 | -2.89 | -1.34 | 0.005 | -0.10 | 0.009 |
| Inv_1_1 | Inv_35_1 | 12.14 | 0.017 | 0.028 | 10.82 | 2.19 | 0.73 | 0.032 | 0.09 | 0.020 |
| Inv_1_1 | Inv_35_2 | 12.14 | 0.017 | 0.018 | 18.56 | -2.37 | -0.91 | 0.027 | -0.08 | 0.031 |
| Inv_25_1 | Inv_7_1 | 10.19 | 0.035 | 0.014 | 14.93 | -2.46 | -0.82 | 0.019 | -0.08 | 0.021 |
| Inv_26_1 | Inv_7_1 | 11.75 | 0.016 | 0.005 | 16.72 | -2.79 | -0.94 | 0.007 | -0.09 | 0.010 |
| Inv_29_2 | Inv_49_1 | 7.50 | 0.023 | 0.008 | 42.41 | 2.67 | 0.98 | 0.009 | 0.11 | 0.006 |
| Inv_35_2 | Inv_43_2 | 19.22 | 0.001 | 0.006 | 20.94 | -2.74 | -1.45 | 0.006 | -0.10 | 0.017 |
| Cross 502 | | | | | | | | | | |
| Inv_22_2 | Inv_47_1 | 12.32 | 0.015 | 0.014 | 19.05 | 2.46 | 0.87 | 0.022 | 0.09 | 0.007 |
| Inv_22_2 | Inv_48_1 | 10.57 | 0.030 | 0.027 | 18.10 | 2.21 | 0.78 | 0.035 | 0.08 | 0.019 |
| Inv_27_1 | Inv_31_2 | 14.46 | 0.005 | 0.001 | 31.34 | -3.39 | -1.32 | 0.001 | -0.10 | 0.002 |
| Inv_27_1 | Inv_32_2 | 9.75 | 0.041 | 0.014 | 24.18 | -2.45 | -0.91 | 0.019 | -0.08 | 0.023 |
| Inv_29_1 | Inv_4_1 | 12.80 | 0.012 | 0.009 | 12.77 | 2.63 | 0.83 | 0.012 | 0.10 | 0.005 |
| Inv_29_2 | Inv_4_2 | 12.80 | 0.012 | 0.013 | 28.15 | 2.50 | 0.98 | 0.017 | 0.10 | 0.009 |
| Inv_31_2 | Inv_54_1 | 14.16 | 0.007 | 0.013 | 16.78 | -2.48 | -0.93 | 0.018 | -0.08 | 0.025 |
| Inv_31_2 | Inv_55_1 | 14.16 | 0.007 | 0.013 | 16.78 | -2.48 | -0.93 | 0.018 | -0.08 | 0.022 |
| Inv_31_2 | Inv_61_2 | 9.81 | 0.045 | 0.005 | 44.21 | 2.78 | 1.06 | 0.012 | 0.12 | 0.004 |
| Inv_35_2 | Inv_45_2 | 12.19 | 0.015 | 0.011 | 34.90 | 2.56 | 1.11 | 0.020 | 0.11 | 0.010 |
| Inv_40_1 | Inv_61_1 | 11.13 | 0.026 | 0.003 | 19.66 | 2.93 | 0.93 | 0.005 | 0.11 | 0.002 |
| Inv_52_1 | Inv_62_2 | 18.28 | 0.001 | 0.002 | 18.36 | -3.10 | -1.22 | 0.003 | -0.08 | 0.004 |
| Cross 541 | | | | | | | | | | |
| Inv_12_2 | Inv_48_1 | 13.53 | 0.008 | 0.024 | 15.00 | -2.26 | -0.84 | 0.030 | -0.07 | 0.034 |
| Inv_28_2 | Inv_43_2 | 12.37 | 0.013 | 0.003 | 43.31 | 2.94 | 1.23 | 0.007 | 0.12 | 0.003 |

*(Continued)*

**Table 2.** (Continued)

| INV_1 | INV_2 | $\chi^2$ _global | $p\_ \chi^2$ _g | $p\_ \chi^2$ _ph | $\chi^2$ _RC | $\chi^2$ _SR | α | α _p | cJ/T | cJ/T_p |
|---|---|---|---|---|---|---|---|---|---|---|
| Inv_29_1 | Inv_8_1 | 12.13 | 0.018 | 0.033 | 9.55 | -2.13 | -0.67 | 0.041 | -0.07 | 0.044 |
| Inv_29_2 | Inv_43_2 | 17.22 | 0.002 | 0.000 | 51.97 | 3.75 | 1.56 | 0.001 | 0.16 | 0.000 |
| Inv_29_2 | Inv_8_1 | 12.13 | 0.018 | 0.023 | 17.57 | 2.27 | 0.84 | 0.028 | 0.08 | 0.017 |
| Inv_29_2 | Inv_9_1 | 10.51 | 0.030 | 0.029 | 18.49 | 2.18 | 0.80 | 0.042 | 0.08 | 0.023 |
| Inv_32_2 | Inv_53_1 | 10.56 | 0.032 | 0.003 | 38.34 | -2.99 | -1.38 | 0.004 | -0.09 | 0.008 |
| Inv_43_1 | Inv_58_1 | 17.52 | 0.001 | 0.021 | 8.49 | 2.31 | 0.73 | 0.028 | 0.09 | 0.015 |
| Inv_43_2 | Inv_58_1 | 17.52 | 0.001 | 0.005 | 20.08 | -2.82 | -1.32 | 0.006 | -0.08 | 0.010 |
| Inv_53_1 | Inv_58_1 | 12.81 | 0.011 | 0.009 | 15.59 | -2.61 | -0.83 | 0.012 | -0.09 | 0.017 |
| Inv_53_1 | Inv_58_2 | 12.81 | 0.011 | 0.005 | 25.19 | 2.81 | 1.01 | 0.007 | 0.11 | 0.003 |
| Cross 62 | | | | | | | | | | |
| Inv_29_1 | Inv_37_2 | 11.10 | 0.024 | 0.043 | 12.58 | 2.02 | 0.92 | 0.048 | 0.10 | 0.034 |
| Inv_29_2 | Inv_37_1 | 11.10 | 0.024 | 0.028 | 18.30 | 2.20 | 1.11 | 0.045 | 0.11 | 0.022 |
| Inv_32_1 | Inv_39_1 | 11.59 | 0.020 | 0.013 | 13.34 | -2.49 | -1.05 | 0.021 | -0.11 | 0.020 |
| Inv_32_1 | Inv_48_1 | 9.95 | 0.036 | 0.033 | 8.02 | 2.13 | 0.96 | 0.047 | 0.09 | 0.027 |
| Inv_32_2 | Inv_39_1 | 11.59 | 0.020 | 0.004 | 24.13 | 2.90 | 1.40 | 0.006 | 0.14 | 0.001 |
| Inv_32_2 | Inv_40_1 | 11.61 | 0.019 | 0.010 | 16.23 | 2.56 | 1.29 | 0.012 | 0.12 | 0.005 |
| Inv_35_2 | Inv_62_1 | 13.71 | 0.005 | 0.000 | 34.49 | -3.64 | -1.88 | 0.000 | -0.14 | 0.002 |
| Inv_49_2 | Inv_5_1 | 11.00 | 0.025 | 0.025 | 17.13 | -2.24 | -1.17 | 0.041 | -0.08 | 0.039 |
| Cross 664 | | | | | | | | | | |
| Inv_24_2 | Inv_43_1 | 10.40 | 0.033 | 0.002 | 36.87 | 3.16 | 1.46 | 0.002 | 0.10 | 0.002 |
| Inv_24_2 | Inv_43_2 | 10.40 | 0.033 | 0.027 | 29.64 | -2.21 | -1.54 | 0.027 | -0.08 | 0.042 |
| Inv_30_1 | Inv_43_2 | 11.08 | 0.026 | 0.025 | 19.37 | -2.25 | -0.91 | 0.034 | -0.07 | 0.034 |
| Inv_32_1 | Inv_43_2 | 11.76 | 0.020 | 0.005 | 24.66 | -2.79 | -1.10 | 0.007 | -0.08 | 0.010 |
| Inv_32_1 | Inv_45_2 | 12.51 | 0.015 | 0.014 | 18.04 | -2.45 | -0.98 | 0.018 | -0.07 | 0.024 |
| Inv_32_1 | Inv_6a_1 | 10.09 | 0.037 | 0.005 | 18.03 | -2.82 | -0.91 | 0.006 | -0.09 | 0.010 |
| Inv_32_2 | Inv_6a_1 | 10.09 | 0.037 | 0.022 | 17.80 | 2.30 | 0.83 | 0.023 | 0.08 | 0.015 |
| Inv_25_1 | Inv_49_2 | 11.85 | 0.017 | 0.002 | 26.71 | -3.06 | -1.20 | 0.004 | -0.09 | 0.006 |
| Inv_49_1 | Inv_45_1 | 10.14 | 0.038 | 0.009 | 13.05 | -2.61 | -0.89 | 0.012 | -0.08 | 0.018 |
| Inv_49_2 | Inv_45_1 | 10.14 | 0.038 | 0.010 | 20.04 | 2.57 | 1.10 | 0.011 | 0.08 | 0.006 |
| Inv_49_2 | Inv_45_2 | 10.14 | 0.038 | 0.015 | 37.22 | -2.43 | -2.14 | 0.016 | -0.09 | 0.029 |

The omnibus $\chi^2$ and *p*-values associated with the 3x3 table are also reported here. $\chi^2$ _**global**: omnibus $\chi^2$ value (3x3 table-derived contingency analysis). ***p*_ $\chi^2$ _g**: uncorrected *p*-value associated with the omnibus $\chi^2$ test. ***p*_ $\chi^2$ _ph**: uncorrected *p*-value associated with the $\chi^2$ *post hoc* test. $\chi^2$ _**RC**: Relative contribution of the contrast to the omnibus $\chi^2$ value (percent). $\chi^2$ _**SR**: Standardized residual of the contrast in relation to the omnibus $\chi^2$ value. **α**: Affinity score. **α _p**: uncorrected *p*-value associated with the affinity score. **cJ/T**: centered Jaccard/Tanimoto score. **cJ/T_p**: uncorrected *p*-value associated to the Jaccard/Tanimoto score.

## Visualization of inversion associations as networks

Networks revealed idiosyncratic patterns of repulsion and attraction; for the same inversion pair, patterns varied from cross to cross. The importance of most inversions within a cross is indicated by node sizes in Fig 4, based on eigenvector centrality measures. While most interactions between edges did not deviate from null expectations (gray edges), there were significant signals of coupling and repulsion according to raw *p*-values derived from the cJ/T index-based hypothesis testing framework. The strongest signals were more abundant in lines 155, 444, 664, and 502, while they were scarcer in the remaining lines.

Network motif analysis revealed significant patterns of inversion interactions across different lines (Fig 5). We found that specific motifs, particularly s.4.4, s.4.6, and s.3.2, were

**Table 3. Example contrasts of the inversion 35 and its dosages states with another inversion for two of the nine analyzed crosses of *Mimulus guttatus*, from lines of Iron Mountain, Oregon, USA.**

| Cross | INV_1 | INV_2 | $\chi^2$_global | $p\_\chi^2$_global | $p\_\chi^2$_post hoc | α | cJ/T |
|---|---|---|---|---|---|---|---|
| C_62 | Inv_35_2 | Inv_62_2 | 13.71 | 0.0054 | 0.04 | 1.04 | 0.113 |
| C_62 | Inv_35_1 | Inv_62_1 | 13.71 | 0.0054 | 0.06 | 0.78 | 0.091 |
| C_62 | Inv_35_1 | Inv_62_2 | 13.71 | 0.0054 | 0.22 | -0.62 | -0.048 |
| C_62 | Inv_35_2 | Inv_62_1 | 13.71 | 0.0054 | 0.00 | -1.88 | -0.141 |
| C_444 | Inv_35_2 | Inv_43_1 | 19.22 | 0.0009 | 0.18 | 0.50 | 0.044 |
| C_444 | Inv_35_1 | Inv_43_1 | 19.22 | 0.0009 | 0.54 | 0.20 | 0.022 |
| C_444 | Inv_35_1 | Inv_43_2 | 19.22 | 0.0009 | 0.45 | -0.29 | -0.025 |
| C_444 | Inv_35_2 | Inv_43_2 | 19.22 | 0.0009 | 0.01 | -1.45 | -0.100 |

The crosses selected were 62 and 444. Blue color indicates coupling interactions, while red color indicates repulsion. $\chi^2$_**global**: omnibus $\chi^2$ value (3x3 table-derived contingency analysis). $p\_\chi^2$_**global**: *p*-value associated with the omnibus $\chi^2$ test. **α**: Affinity score. **cJ/T**: centered Jaccard/Tanimoto score.

consistently enriched across multiple crosses, with observed/expected ratios close to or exceeding 1.5. The strongest signals were detected in crosses 502 and 909, where certain motifs showed high statistical significance (-log10(adj. P) > 1.25) and substantial enrichment compared to random expectations. Z-score analysis (Fig 5D) further supported the non-random distribution of these motifs, with several network-motif pairs showing significant deviations from expected frequencies. The relationship between observed and expected motif counts demonstrated clear structural patterns in the inversion co-occurrence networks, suggesting organized rather than random associations between inversions (Fig 5E).

The most prominent result derived from the estimated network was the variability in coupling/repulsion patterns. In some cases, a pair of inversions interacted positively in one cross but negatively in another. As an example, we examined inversions 29, 32, and 40, which appeared in all studied lines. There was no consistent relationship between nodes in the networks across lines. In all cases, inversion dosage combinations exhibited different patterns of repulsion and attraction as a function of the genetic family – cross (Fig 6).

This finding was further supported by a formal replicated independence G test, in which we found no significant pooled effect (Table 4), suggesting fluctuating relationships between inversions across lines. Additionally, we found evidence of heterogeneity for one of the analyzed contrasts. In this case, the relationship between inversion 40 in heterozygous state and inversion 32 in homozygous state appeared to differ significantly among genetic families (crosses).

Interestingly, we found no relationship between single inversion distortion and inter-inversion interaction. The SDV, a measure directly associated with the effect size of segregation distortion, was not significantly associated with the number of significant tests with raw *p* values less than 0.05 where the distorted inversion appears ($t = 0.66$, p = 0.507). Additionally, Fig 7 shows a weak association between these two measures and between the mean *p* value of the distorted inversions and their own SDV.

## Discussion

Genomic inversions are structural mutations that reverse the orientation of chromosomal segments, reducing the recombination rate in heterozygotes The suppression of recombination acts as an effective barrier to genetic exchange between different orientations, except for cases of gene conversion or 'gene flux' [18,41]. Despite recent interest in the evolutionary role of inversions [42–44], understanding of these and other structural variants remains limited,

although they are increasingly recognized as a substantial source of genetic variability in natural populations [45,46]. Inversions can also be central elements in reproductive isolation, given their major effect as recombination suppressors [20]. Many studies in the last decade have focused on understanding the role of inversions in phenotypic polymorphisms [47], behavior [48], the evolution of sex chromosomes [49], and other phenomena [46].

Traditionally, inversions have been considered in isolation, even though they are more common within populations than previously thought. In many species, single individuals may carry inversions at multiple *loci* [19,50], making it imperative to consider the interaction networks among inversions. Co-occurrence analysis offers a powerful approach to unravel potential epistatic relationships between inversions. The study of epistatic interactions between inversion *loci* could shed light on diverse phenomena. For instance, inversion interactions could shape hybrid incompatibilities, as minor parent ancestry is less likely to persist in genome regions with low rates of hybridization [51]. An inversion can prevent gene flow between populations at the genetic *loci* contained within it, consequently preventing deleterious reshuffling of alleles between local and migrant entities [52]. From a more global perspective, we can explore the degree of coadaptation of a given set of inversions [15]. For the data considered in this paper, with plants produced from controlled crosses, gametic competition or the differential germination/survival of F2 plants must be the cause of co-occurrence, which makes the interpretation of results straightforward.

As a key finding of this research, we observed a significant impact of specific inversion groups on plant survival, using segregation distortion values of individual inversions as a proxy. Among the significant signals detected, most inversions were cross-specific, though inversion 53 showed common distortion across three crosses, suggesting a moderately pervasive effect in the population. These observations of significant departures from Mendelian inheritance across multiple chromosomal inversions provide compelling evidence for investigating genotypic co-occurrence patterns in this system. Our sequential $\chi^2$ testing approach revealed consistent signatures of zygotic selection rather than gametic selection, indicating that selective pressures act on specific genotype combinations after fertilization. This is particularly evident in inversions such as Inv_49, Inv_55, and Inv_53, which exhibit highly significant G-statistics and substantial segregation distortion values.

The observed genotypic frequency patterns demonstrate strong asymmetries, with certain genotype combinations consistently underrepresented across different inversions. For instance, Inv_49 shows a striking disparity between RR (92) and II (0) genotypes, suggesting strong selective pressures against specific genotypic states. These patterns, combined with significant zygotic selection signatures, indicate that the inversions are not evolving independently but are subject to functional interactions influencing their co-occurrence. Selection strength varies among inversions, as evidenced by segregation distortion values ranging from 114.38 to 2.2. This variation suggests a complex selective landscape where different inversions experience varying degrees of selective pressure. Such heterogeneity in selection intensity further supports the need for genotypic-based co-occurrence analysis, as it could reveal whether inversions under stronger selection tend to show specific association patterns with other inversions.

Given these observations, genotypic-based co-occurrence analysis becomes crucial for understanding the evolutionary dynamics of these inversions in current evolutionary studies. This approach could reveal potential epistatic interactions between inversions, identify functionally related inversion modules, and uncover possible developmental constraints influencing which inversion combinations can coexist. Furthermore, such analysis could provide insights into how these inversions might contribute to adaptation, particularly if certain inversion combinations tend to be maintained together more frequently than expected by chance.

The clear evidence for zygotic selection also suggests these inversions might play important roles in development or fitness-related traits. Understanding their genotypic co-occurrence could therefore provide valuable insights into the functional architecture of adaptation in this system. This knowledge is particularly relevant for understanding how structural variants like inversions contribute to evolutionary processes and genetic variation maintenance in natural populations.

## The utility of co-occurrence metrics

Applied to this study, co-occurrence metrics reveal patterns of putative coupling (positive epistasis) and repulsion (incompatibilities) detectable within a genotypic scope. Our simulations demonstrate the advantages of these metrics, which have been primarily used in ecology. Traditional tests analyze the full 3-by-3 genotype table (Fig 1), whereas Affinity and cJ/T metrics operate on "slices" of this table. An omnibus test provides greater power in detecting overall signals related to expectation deviations. Co-occurrence metrics offer more specificity, providing granular resolution of the underlying causes of such deviations. Fig 3 illustrates the practical potential of these indexes. When simultaneous inflation of one category and deflation of another occur, omnibus tests conflate these effects. In contrast, the affinity index and Jaccard/Tanimoto measures detect distinct signals and identify interactions as either positive or negative.

When applied to the *Mimulus* data, co-occurrence metrics reveal a pattern of variable epistasis (Table 2), with network representation providing a clear picture of this complexity (Fig 4). This depiction encompasses all inversions present in each line and all valid interactions established between *loci*. The architecture of these networks is cross-specific, and the importance of each inversion varies across crosses. Interactions between inversion pairs are modified by the overall genetic background. Almost every set of inversions in each genetic family is unique, consequently altering the association network as a function of the present inversion subset. These findings align with the concept explored by [53] of cryptic genetic variation and how background-dependent effects may be more prevalent than our and other current findings suggest. This concept fundamentally posits that a mutation's effect on the organismal phenotype space (survival in this study) depends not only on the focal component associated with it (inversions and their interactions) but also on the interplay and epistatic relationships established in the genetic background space and intermediate phenotype level (gene expression level).

Overall, the deviation patterns indicate that heterozygous-homozygous combinations exhibit the highest number of significant deviations from independence. However, we cannot directly associate this result with a determined expectation since, due to our experimental design, we observed only the immediate product of selection in F2 plants in terms of survival. We lacked prior information about how specific genotypic combinations between inversions might have shaped the survivorship space of *M. guttatus*. Notably, none of the deviations displayed an unequivocal directional bias, meaning that coupling and repulsion interactions occurred at equivalent frequencies.

The network motif analysis revealed consistent patterns of inversion interactions that provide insights into the structural organization of chromosomal inversions in *M. guttatus*. Specifically, motifs s.4.6 and s.3.2 (Fig 5) showed significant enrichment across multiple lines, suggesting these association patterns are not random but likely reflect underlying biological constraints or advantages. In the context of chromosomal inversions, these undirected motifs can represent different scenarios of genetic architecture integration. For instance, highly prevalent patterns (as observed in crosses 909 and 502) may indicate the group sizes and overall

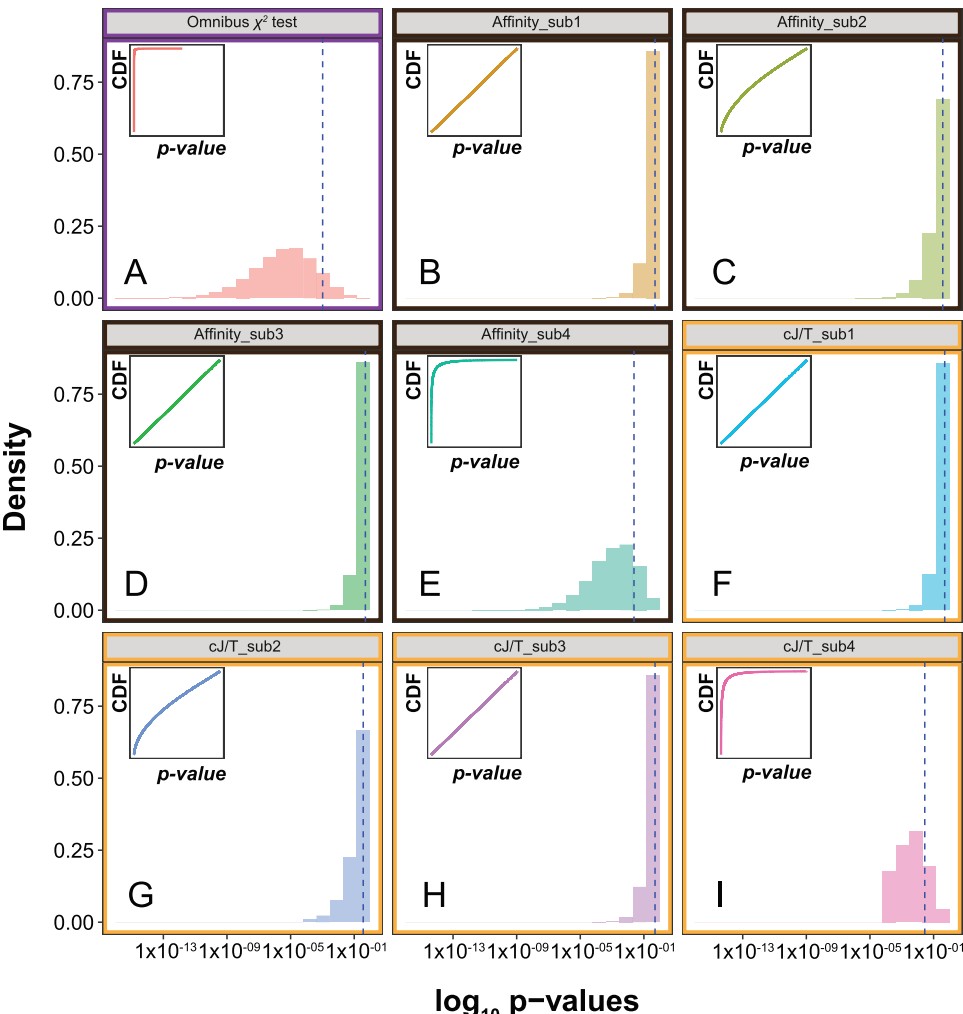

**Fig 3. Distribution of the *p*-values derived from the omnibus $\chi^2$ test, the affinity analysis, and the centered Jaccard/Tanimoto-based analysis under a situation with a deflation in the counts of the 8th cell and an inflation of those in the 9th cell.** Those coordinates are based on Fig 1B. There is a shift in the distribution of the *p*-values to the left in the case of the omnibus $\chi^2$ test (both signals are merged), while in the cases of the submatrices 2 and 4 analyzed with the affinity and Jaccard/Tanimoto indexes, both events are captured as deviations from the null model (Affinity_sub2, Affinity_sub4, cJ/T_sub2, and cJ/T_sub4). Histograms showing the density of each bin for the log-transformed *p*-values are complemented with a subplot depicting the cumulative distribution function (CDF).

structural shape of interaction networks between inversions that must be inherited together to maintain adaptive allele combinations or prevent deleterious genetic combinations. The variation in motif enrichment across lines further suggests that these association patterns are influenced by the genetic background, consistent with previous results. Importantly, the detection of significant motifs, even without considering the direction of interactions (coupling or repulsion), indicates fundamental organizational principles in how inversions can be combined within a genome. This network-based analysis provides a robust method for exploring multiway interactions that extends beyond the classic pairwise tests traditionally used in linkage disequilibrium and association studies [54].

From all pairwise contrasts, we extracted a case study featuring three inversions (29, 32, and 42) that segregate in all nine crosses (Fig 6). The results clearly demonstrate how specific

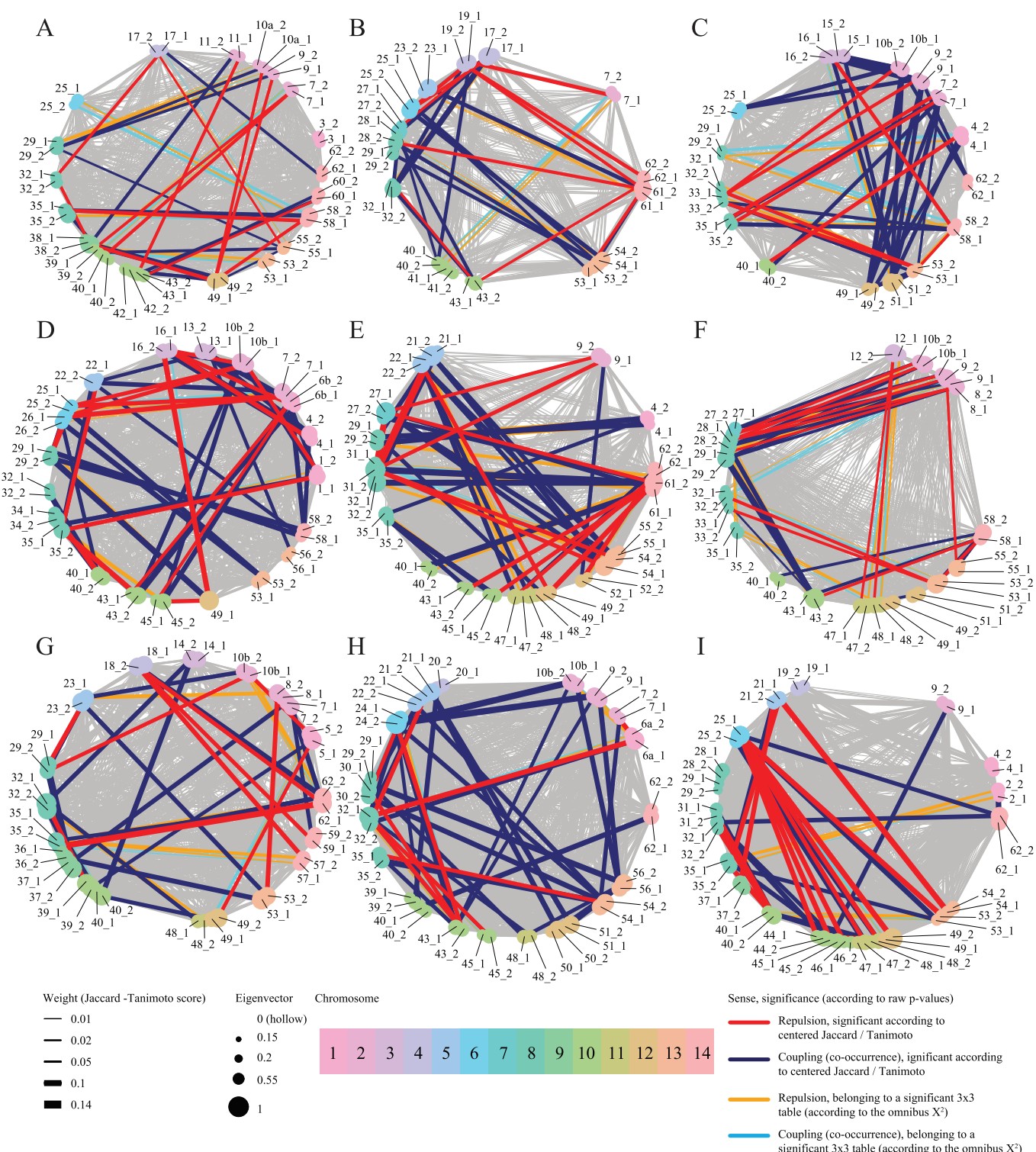

**Fig 4. Networks based on the Jaccard/Tanimoto score for nine genetic families of *Mimulus guttatus* from Iron Mountain, Oregon, USA.** Significant results are those detected by all three 2x2 table-based approaches. In gray are represented the edges that were not recognized by the omnibus $\chi^2$ test. A) C_1034. B) C_1192. C) C_155. D) C_444. E) C_502. F) C_541. G) C_62. H) C_664. I) C_909.

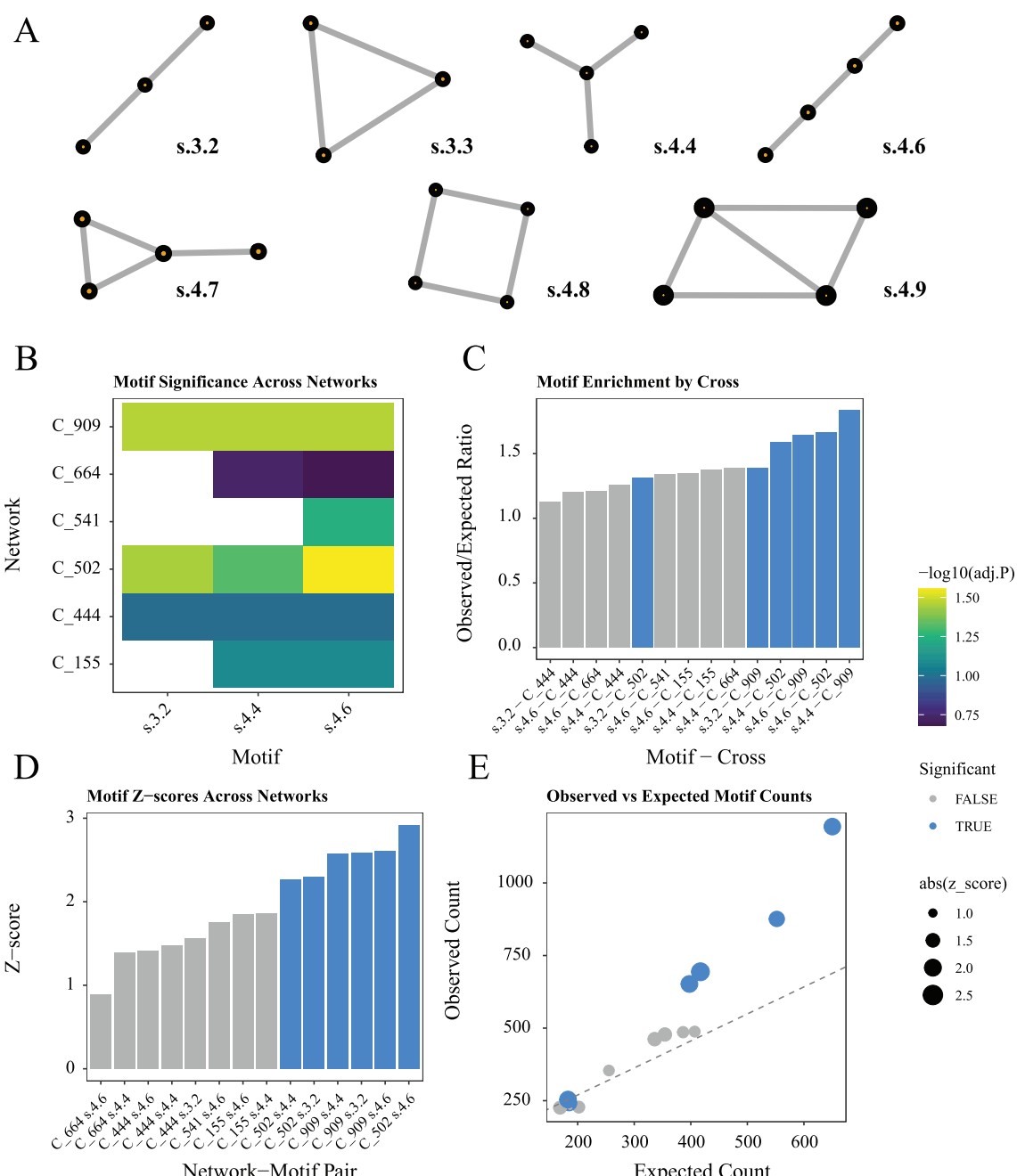

**Fig 5. Systematic analysis of network motifs in inversion co-occurrence networks across six *Mimulus guttatus* lines from Iron Mountain, Oregon, USA.** (A) Third and fourth order motifs. (B) Statistical significance of different motifs across networks shown as -log10 of adjusted *p*-values. Tiles in blank correspond to motif-cross combination excluded from analysis with observed counts below 200. (C) Enrichment analysis showing the ratio of observed to expected motif frequencies for each line-motif combination. (D) Z-score distribution for network-motif pairs, indicating deviation from random expectation. (E) Scatter plot of observed versus expected motif counts; point size indicates absolute Z-score and color denotes statistical significance (TRUE/FALSE, FDR < 0.05). Crosses analyzed: C_155, C_444, C_502, C_541, C_664, and C_909. Motifs analyzed include s.3.2, s.4.4, and s.4.6, representing different patterns of inversion co-occurrence.

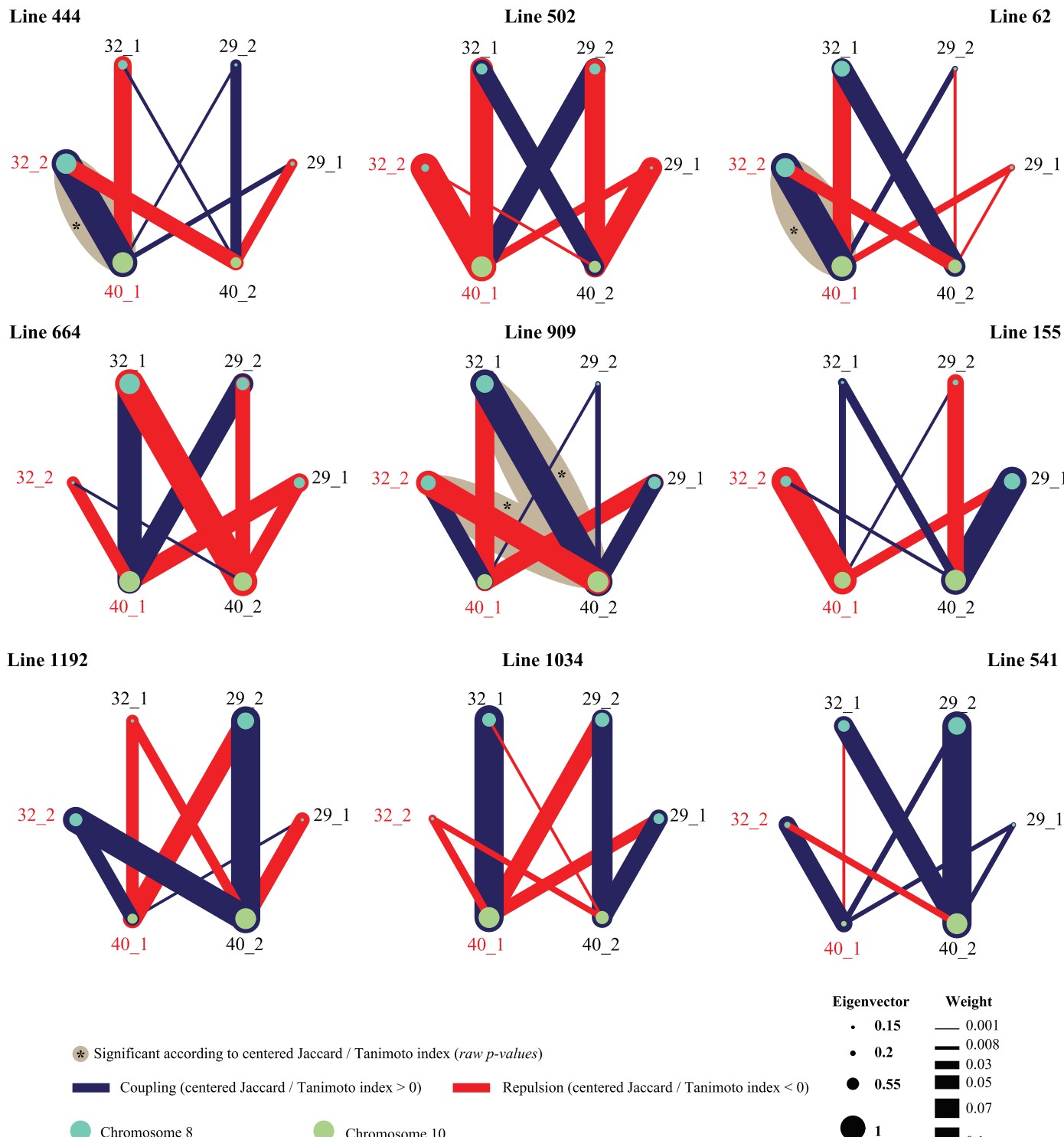

**Fig 6. Co-occurrence network for the homozygous and heterozygous states of inversions 29, 32, and 40 in nine crosses of a population of *Mimulus guttatus*, Iron Mountain, Oregon, USA.** In blue are depicted the attraction interactions, while in red are depicted the repulsion ones. The size of the nodes is based on the eigenvectors (measure of importance in the network). A star symbol signals those interactions that are significant according to the centered Jaccard/Tanimoto. Red labels correspond to inversions which interaction pattern was detected as heterogeneous through a replicated independence G-test ($G_{het}$ = 17.766, $p$ = 0.023).

**Table 4. Replicated G test of independence between inversion dosages present in all nine crosses performed using *Mimulus guttatus* lines from a natural population of Iron Mountain, Oregon, USA.**

| Inv_1 | Inv_2 | $G_{total}$ | $df_{total}$ | $p_{total}$ | $G_{pooled}$ | $df_{pooled}$ | $p_{pooled}$ | $G_{het}$ | $df_{het}$ | $p_{het}$ |
|---|---|---|---|---|---|---|---|---|---|---|
| Inv_40_1 | Inv_29_1 | 4.448 | 9 | 0.880 | 1.473 | 1 | 0.225 | 2.975 | 8 | 0.936 |
| Inv_40_2 | Inv_29_1 | 7.114 | 9 | 0.625 | 0.041 | 1 | 0.839 | 7.072 | 8 | 0.529 |
| Inv_40_1 | Inv_29_2 | 3.326 | 9 | 0.950 | 0.142 | 1 | 0.706 | 3.184 | 8 | 0.922 |
| Inv_40_2 | Inv_29_2 | 5.827 | 9 | 0.757 | 0.592 | 1 | 0.441 | 5.235 | 8 | 0.732 |
| Inv_40_1 | Inv_32_1 | 12.057 | 9 | 0.210 | 1.364 | 1 | 0.243 | 10.694 | 8 | 0.220 |
| Inv_40_2 | Inv_32_1 | 13.558 | 9 | 0.139 | 2.052 | 1 | 0.152 | 11.506 | 8 | 0.175 |
| Inv_40_1 | Inv_32_2 | 20.229 | 9 | 0.017 | 2.464 | 1 | 0.117 | 17.766 | 8 | 0.023 |
| Inv_40_2 | Inv_32_2 | 10.205 | 9 | 0.334 | 1.465 | 1 | 0.226 | 8.740 | 8 | 0.365 |

**INV_1** and **INV_2** refer to the two inversion dosages considered in an independence test.

$G_{total}$, $df_{total}$, and $p_{total}$ are the statistic, degrees of freedom, and *p*-value associated to the total G test. $G_{pooled}$, $df_{pooled}$, and $p_{pooled}$ are the statistic, degrees of freedom, and *p*-value associated to the pooled G test. $G_{het}$, $df_{het}$, and $p_{het}$ are the statistic, degrees of freedom, and *p*-value associated to the heterogeneity G test.

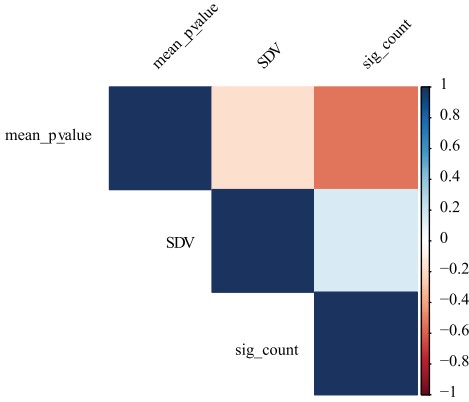

**Fig 7. Correlation space between the segregation distortion value of the inversion (SDV), the number of times the inversion is implicated in a pairwise contrast with a raw *p* value less than 0.05, and the average *p* value of that inversion across all valid pairwise contrasts.** The inversions studied are from a natural population of *Mimulus guttatus from* Iron Mountain, Oregon, USA. The correlation coefficients are derived from the Spearman test (rank-based approach).

genotype interactions vary between crosses, thus illustrating how selection changes as a function of the genetic background. The replicated independence test confirms the significance of this variation (Table 4), while the Jaccard/Tanimoto metrics quantify the nature of coupling (positive) and incompatible (negative) interactions. Regarding the pooled effect, we observed no significant signal—there is no directional epistasis. In network terminology, there are no conserved edges. The study identified two key inversions for which the heterogeneity test revealed significant effects: inversion 32 (in homozygous state) and inversion 40 (in heterozygous state). This suggests that epistatic interactions between those inversions are not consistent across different genetic backgrounds. This indicates that a particular combination of inversions might be advantageous in one genetic context but disadvantageous in another. Therefore, when evaluating the selective advantage of inversions, it is crucial to consider how they interact within the entire genetic system rather than studying them in isolation [15].

Importantly, inversions showing segregation distortion were not more or less likely to interact with other inversions. The association between the SDV, a measure of each inversion's individual effect, and the association relationships was minimal (Fig 7). The SDV correlates weakly with both the number of pairwise contrasts in which that inversion is implicated with a raw *p* value under 0.05 and the mean *p* value of such pairwise contrasts. We had anticipated observing a positive association regarding the number of strong interactions, assuming that a highly distorted *loci* might be more implicated in coupling and repulsion within the genomic context. This addresses an important question in QTL mapping: are regions showing a marginal (single locus) effect more likely to exhibit interactions with other parts of the genome? Our results are negative in this regard. This decoupling of individual and interactive effects aligns with modern perspectives on the modular nature of genetic architectures (Wagner *et al.*, 2007) and the complex relationship between individual allelic effects and epistatic networks [55].

When a particular inversion at a given dosage level occurs more frequently alongside another inversion at a specific dosage level, we can conceptualize this relationship as a putative functional interaction. Patterns of epistatic interactions in the context of inversions have been previously identified using linkage disequilibrium-based approaches. For instance, results from [56] suggest that inversions in *Drosophila pseudoobscura* contribute to maintaining positive epistatic interactions between *loci* included in individual inversions. However, according to our literature review, findings regarding epistatic interactions strictly between inversions (functional association between genes from different inversions) are rare, particularly in plants.

Most research in this area has been conducted with *Drosophila* spp. [15]. Among studies investigating linkage disequilibrium between inversions, [57] reported interactions between different inversions in *Drosophila subobscura*, although evidence only supported intrachromosomal interactions. This limitation might stem from methodological constraints, as epistatic interactions between unlinked inversions in different chromosomes have been documented in *Drosophila melanogaster* [58].

## Future directions

Our study demonstrates the potential of co-occurrence analysis in unraveling the complex interactions between inversions. This approach offers a novel perspective on several key aspects of inversion biology:

**1) Capturing complex genetic interactions.** Co-occurrence analysis provides an approach to understanding genetic systems [57]. Directed network analysis of motifs in the observed networks could also provide additional information about the directional nature of regulatory structures, improving the understanding of how these structures shape the association of genetic *loci*. This particular aspect is an active effort in bioinformatics, given that is a challenging problem which complexity increase with the number of nodes under consideration [36].

**2) Identifying potential supergene associations.** Co-occurrence analysis could help identify interactions between supergenes by revealing consistent patterns of association between specific inversions and gene clusters. Since supergenes consist of coadapted genes locked within genomic inversions and inherited as single units [59], persistent co-occurrence patterns between different supergenes could indicate functional relationships or selective advantages of specific supergene combinations. This could reveal how supergenes work together at a broader genomic level.

**3) Exploring inversion dynamics.** Our approach provides insights into the evolutionary dynamics of genomic inversions across different chromosomal locations. While inversions are known to suppress recombination between arrangements when in heterozygous state

[41,60,61], our co-occurrence analysis focuses on a different scale - examining how different inversion loci are inherited together across the genome. This allows us to understand potential functional relationships between inversions and their collective role in the evolutionary history of the species[45,46].

**4) Refining our understanding of inversion evolution.** Despite recent interest in the evolutionary role of inversions [42], our understanding remains limited. Co-occurrence analysis offers a new tool to explore how inversions evolve and interact, potentially revealing patterns that other methods might miss. Future research should aim to expand our approach, potentially integrating co-occurrence analysis with other genomic and phenotypic data. This could provide a more comprehensive understanding of inversion effects and their role in evolution. Additionally, applying this method to larger datasets and diverse species could reveal broader patterns in inversion dynamics and evolution.

## Conclusions

This study demonstrates the application of co-occurrence statistics—traditionally used in community ecology—to investigate interactions between chromosomal inversions in *Mimulus guttatus*. By employing multiple statistical approaches, including contingency table analysis, centered Jaccard/Tanimoto index, and affinity scores, we detected patterns of coupling and repulsion between inversions across different experimental crosses. Our findings reveal a complex landscape of inversion interactions, with some pairs co-occurring more frequently than expected by chance among F2 plants, while others exhibit patterns of repulsion. These interactions varied across different genetic backgrounds, highlighting the context-dependent nature of inversion effects.

The network analysis approach provided a novel visualization of these complex interactions, revealing line-specific patterns and centrality of certain inversions within the genomic architecture. This method offers a promising tool for future studies aiming to unravel the functional relationships between structural variants in the genome. Our study also underscores the importance of considering dosage effects in inversion studies, as we observed different interaction patterns for homozygous and heterozygous states of inversions. This nuanced approach provides a more comprehensive understanding of how inversions may influence fitness and adaptation.

While our research provides valuable insights into inversion dynamics in *M. guttatus*, it also highlights the need for larger sample sizes and more comprehensive genomic data to fully capture the complexity of these interactions. Future studies should aim to integrate this co-occurrence approach with phenotypic data and functional genomics to elucidate the mechanisms underlying these interaction patterns. In conclusion, this study demonstrates the utility of ecological co-occurrence methods in genomics research and opens new avenues for investigating the role of structural variants in evolution. By bridging concepts from ecology and genetics, we provide a framework for future research on the complex interplay between genomic architecture and adaptive evolution.

## Supporting information

**S1 Fig. Distribution of the *p*-values derived from the omnibus $\chi^2$ test, the affinity analysis, and the centered Jaccard/Tanimoto-based analysis under the null model.** Histograms showing the density of each bin for the log-transformed *p*-values are complemented with a subplot depicting the cumulative distribution function (CDF).
(EPS)

**S2 Fig. Distribution of the *p*-values derived from the omnibus $\chi^2$ test, the affinity analysis, and the Jaccard/Tanimoto-based analysis under a situation with a deflation of the 9th cell counts.** Those coordinates are based on Fig 1A. There is a shift in the distribution of the *p*-values to the left, given the deviation that this modification implies for the null hypothesis. Histograms showing the density of each bin for the log-transformed *p*-values are complemented with a subplot depicting the cumulative distribution function (CDF).
(EPS)

**S3 Fig. Linear Associations Among Various Metrics. This figure illustrates the linear relationships between different pairs of metrics.** Plot A shows the relation between the standardized residuals from the $\chi^2$ *post hoc* test and the centered Jaccard/Tanimoto scores. Plot B depicts the relationship between the standardized residuals from the $\chi^2$ *post hoc* test and the affinity score. Lastly, Plot C demonstrates the linear relationship between the centered Jaccard/Tanimoto score and the affinity score. Each plot includes a trend line, indicating the direction and strength of the linear relationship between the two metrics. The summary of the model is included in the superior edge of each subplot.
(EPS)

**S4 Fig. Mean value variation of various metrics across the allele frequency landscape defined by two hypothetical inversions.** Uppermost left panel, *D*. Uppermost right panel, *D'*. Bottom left panel, *cJ/T*. Bottom right panel, *α*.
(EPS)

**S1 Word Doc. Extended methods.**
(DOCX)

**S2 Word Doc. Supporting Information.**
(DOCX)

**S1 Excel Spreadsheet. Plant genotypes for each of the inversions analyzed in this study.**
(XLSX)

## Acknowledgements

We thank the ongoing *Mimulus* pan-genome project for the early access to IM767 and IM62 genomes, specially John Willis and Lila Fishman. Also, we thank the journal editor and the reviewers for their valuable feedback and recommendations, which significantly improved the content and the way we convey the information in the present paper.

## Author contributions

**Conceptualization:** Luis Madrigal-Roca, John K. Kelly.

**Data curation:** Luis Madrigal-Roca, John K. Kelly.

**Formal analysis:** Luis Madrigal-Roca, John K. Kelly.

**Funding acquisition:** John K. Kelly.

**Investigation:** Luis Madrigal-Roca, John K. Kelly.

**Methodology:** Luis Madrigal-Roca, John K. Kelly.

**Project administration:** Luis Madrigal-Roca.

**Resources:** Luis Madrigal-Roca.

**Software:** Luis Madrigal-Roca.

**Supervision:** John K. Kelly.

**Validation:** Luis Madrigal-Roca, John K. Kelly.

**Visualization:** Luis Madrigal-Roca.

**Writing – original draft:** Luis Madrigal-Roca, John K. Kelly.

**Writing – review & editing:** Luis Madrigal-Roca, John K. Kelly.

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
