## [Decision Letter · Decision Letter 0]

29 Dec 2024

PONE-D-24-52521Are you with me? Co-occurrence tests from community ecology can identify positive and negative epistasis between inversions in Mimulus guttatusPLOS ONE

Dear Dr. Madrigal-Roca,

Thank you for submitting your manuscript to PLOS ONE. After careful consideration, we feel that it has merit but does not fully meet PLOS ONE’s publication criteria as it currently stands. Therefore, we invite you to submit a revised version of the manuscript that addresses the points raised during the review process.

**The reviewers were generally excited about the approch introduced in the manuscript. The main points for improvement are the following.**

**Provide better explanation for the metrics, as they are novel to the PopGen community. Can be done with extended methods, or supplemental methods. In the intro, the network is presented as an alternative approach to the cooccurance stats, but here it is used as a follow up analyses/visualization.**

**Elaborate on strenghts/weaknesses of these methods to detect coupling vs repulsion (suggest using the LD metric vocabulary).**

**Can you compare the preformance of the metircs with established estimators, like linkage disequilibrium (D and D’). We are not ask for full scale benchmark analyses, but comparing the outputs of the different metrics can illustrate the nature of the signals they are best equipped to identify. Mention in disussion that benchmark analyses on simulated data with various properties (missingness, patterns of coupling, repulsion, recombination rates etc) would be an idean next step.**

**Can you synchronize the terminology and labelling of the different methods? Like between Figure 1 and 2, and text for the simulation methods, it is a bit hard to follow which method is which. Perhaps it is just an issue of improving Figure 2 and its legend ? (dont need to state “P-values“ in heading of all 9 panels)**

**Provide a well annotated R script (github ideally) that explains the steps in the analyses, with well described (metadata) dataset. This will increase likelihood of the methods being picked up by other groups.**

**If the cooccurance patterns depend strongly on Cross, then you are reporting higher order interactions. Cite literature on those (rather than organism specific papers), like https://www.sciencedirect.com/science/article/pii/S0168952513000218?via%3Dihub**

**Intermediate**

**Think it may be helpful to generate a figure for the data on 336-242, to demonstrate how the numbers look for the 3x3 tables. This may provide a better link to the graphics following! Your call!**

**Check wording and grammar of entire manuscript, see our minor points!**

We look forward to receiving your revised manuscript.

Kind regards,

Arnar Palsson, Ph.D.

Academic Editor

PLOS ONE

**Journal Requirements:**

1. When submitting your revision, we need you to address these additional requirements. Please ensure that your manuscript meets PLOS ONE's style requirements, including those for file naming. The PLOS ONE style templates can be found at https://journals.plos.org/plosone/s/file?id=wjVg/PLOSOne_formatting_sample_main_body.pdf and https://journals.plos.org/plosone/s/file?id=ba62/PLOSOne_formatting_sample_title_authors_affiliations.pdf 2. We note that the grant information you provided in the ‘Funding Information’ and ‘Financial Disclosure’ sections do not match.  When you resubmit, please ensure that you provide the correct grant numbers for the awards you received for your study in the ‘Funding Information’ section. 3. Thank you for stating the following financial disclosure: National Institute of General Medical Sciences (NIGMS/NIH) under award number P30GM145499. Please state what role the funders took in the study.  If the funders had no role, please state: "The funders had no role in study design, data collection and analysis, decision to publish, or preparation of the manuscript." If this statement is not correct you must amend it as needed. Please include this amended Role of Funder statement in your cover letter; we will change the online submission form on your behalf. 4. Please note that your Data Availability Statement is currently missing the repository name. If your manuscript is accepted for publication, you will be asked to provide these details on a very short timeline. We therefore suggest that you provide this information now, though we will not hold up the peer review process if you are unable. 5. When completing the data availability statement of the submission form, you indicated that you will make your data available on acceptance. We strongly recommend all authors decide on a data sharing plan before acceptance, as the process can be lengthy and hold up publication timelines. Please note that, though access restrictions are acceptable now, your entire data will need to be made freely accessible if your manuscript is accepted for publication. This policy applies to all data except where public deposition would breach compliance with the protocol approved by your research ethics board. If you are unable to adhere to our open data policy, please kindly revise your statement to explain your reasoning and we will seek the editor's input on an exemption. Please be assured that, once you have provided your new statement, the assessment of your exemption will not hold up the peer review process. 6. Please amend either the abstract on the online submission form (via Edit Submission) or the abstract in the manuscript so that they are identical. 7. Please upload a new copy of Figure 3 as the detail is not clear. Please follow the link for more information: https://blogs.plos.org/plos/2019/06/looking-good-tips-for-creating-your-plos-figures-graphics/"
https://blogs.plos.org/plos/2019/06/looking-good-tips-for-creating-your-plos-figures-graphics/ **Additional Editor Comments:**

PLOS24_Inversions

The reviewers were generally excited about the approch introduced in the manuscript. The main points for improvement are the following.

1. Provide better explanation for the metrics, as they are novel to the PopGen community. Can be done with extended methods, or supplemental methods. In the intro, the network is presented as an alternative approach to the cooccurance stats, but here it is used as a follow up analyses/visualization.

2. Elaborate on strenghts/weaknesses of these methods to detect coupling vs repulsion (suggest using the LD metric vocabulary).

3. Can you compare the preformance of the metircs with established estimators, like linkage disequilibrium (D and D’). We are not ask for full scale benchmark analyses, but comparing the outputs of the different metrics can illustrate the nature of the signals they are best equipped to identify. Mention in disussion that benchmark analyses on simulated data with various properties (missingness, patterns of coupling, repulsion, recombination rates etc) would be an idean next step.

4. Can you synchronize the terminology and labelling of the different methods? Like between Figure 1 and 2, and text for the simulation methods, it is a bit hard to follow which method is which. Perhaps it is just an issue of improving Figure 2 and its legend ? (dont need to state “P-values“ in heading of all 9 panels)

5. Provide a well annotated R script (github ideally) that explains the steps in the analyses, with well described (metadata) dataset. This will increase likelihood of the methods being picked up by other groups.

6. If the cooccurance patterns depend strongly on Cross, then you are reporting higher order interactions. Cite literature on those (rather than organism specific papers), like https://www.sciencedirect.com/science/article/pii/S0168952513000218?via%3Dihub

Intermediate

Think it may be helpful to generate a figure for the data on 336-242, to demonstrate how the numbers look for the 3x3 tables. This may provide a better link to the graphics following! Your call!

Minor points.

Line 45

Don’t agree with the last part of sentence “In genetics, non-random relationships between loci, described as linkage disequilibria, are measures of co-occurrence”. The non random relationships are not measures!

Line 49.

Can you reword “Attraction or repulsion of species in their location”

Line 58

Metrics?? “The co-occurrence techniques”

Line 64

Drop “of” “In our study species of Mimulus guttatus”

Line 77

Provide references ”Network analysis is an area of active development in biology.”

Line 79

Singular or plural idea(s)? “The underlaying basic ideas”

Line 104.

Which 10 genomes? “The ten genomes were derived”

Line 130.

“if genotype frequencies” these are all inversion genotypes?

Line 140

Reword, “independence between the pairwise patterns of dosage between inversions” maybe drop “dosage”?

Line 188

Where does this “survival rate” concept come from? Define it above.

Line 205

How many datapoints were in this comparison dataset? “also detect signals derived from linkage.”

Line 211

Used rather than adjusted? “we adjusted robust linear regression instead”

Line 247-251.

Reword this section, introduce the permutation approach earlier in the description.

Line 258.

“in six other crosses where it displays no distortion.” Or state directly, “where both types of homozygotes are seen”?

Line 274

Table 1. Can you annotate the p values in different way? X21 followd by column with its p valunes, and then X22? And a column that has all the same values is useless, i.e. “Selection”

Line 276.

Maybe reword title to “Testing of methods via simulation(s??)”?

Line 286.

Maybe say “counts” were “deflated”, not cell?

Line 309.

“Of the valid contrasts…” sentence is convoluted. Can this be subdivided into 2 or more sentences or reworded?

Line 318

Check refereence to supplement II.

Line 320

Any cases of the opposite?

“but the” instead of “and the”?

Line 366

Was there good or poor congruence? Consider changing the section heading “Congruence between methods” or first sentence “The congruence between the 2x2 table-based methods is depicted in Fig. 3 and Fig. 3S.” to better explain the main result of this section.

Line 371.

“robust”, is this word needed? Anything specificly robust about the regression?

Line 367 and Figure 3.

It is unclear what is shown in the figure (legend missing details, what is above and below diagonal. Labels on figure to small to read, question if inset colour panel is needed for all 9? Make start of Figure 3 legend more transparent “good correspondance was betw…”

Line 384.

Isnt this more of a visualization of the results from previous section??

Heading “Visualization of co-occurance as inversion networks” or some version of this?

Line 390

Quesiton of wording ”there were significant patterns”, maybe “signal”?

Line 391

“Strongest signals were in lines 155”

Line 439

“individuals” rather than “organisms”

Line 442.

Wording, “Also, inversions can be central elements in our understanding about”, maybe drop the “our understanding”?

Line 453.

“interactome” is not the right word here, find another. Also fix in other places in MS.

Line 456

“induced” another verb may be better!

Line 470

Reword “although often not across all crosses” was it not so that this was mainly in few crosses?

Line 479-

Check sentence structure and placement of reference.

Line 482

Strange wording “…where the higher count of plants was found in the homozygous”. Can you rephrase. “more plants were homozygous for inversion 49 …”?

Line 485

“effective population size”?

Line 491

Wording? “Applied to in this study”

Line 507

Add citations to papers about higher order epistasis, see above.

Line 511

“We investigated one such example in detail” rather than “Figure 5 provides a particular example of variability of interactions.” Or some other more descriptive version of this

Line 518-

Reword, in particular klonky text at “In the case of the inversion 32 in homozygous state and the inversion 40 in heterozygous state, it can be seen a significant effect for the heterogeneity”

Line 528

“strong” not “string”?

Line 530

Verb missing…”are?”

Line 533

Maybe we just need more power, or that the dependcies of SDV and inversion interactions are diverse, on case by case basis?

Line 565 and 570

Add references to these sections, so to bring attention to these methods through scholar working on these topics.

Line 579

Or how inversion combinations are depleted or enriched in hybrids between species or subspecies.

Line 584.

Can also be used to study combinations of specific alleles of the genes within inversions, both within one inversion or across inversions.

Line 628

Wording ”in special to” ?

Reviewers' comments:

Reviewer's Responses to Questions

**Comments to the Author**

1. Is the manuscript technically sound, and do the data support the conclusions?

Reviewer #1: Partly

Reviewer #2: Yes

2. Has the statistical analysis been performed appropriately and rigorously? 

Reviewer #1: Yes

Reviewer #2: Yes

3. Have the authors made all data underlying the findings in their manuscript fully available?

Reviewer #1: Yes

Reviewer #2: Yes

4. Is the manuscript presented in an intelligible fashion and written in standard English?

Reviewer #1: Yes

Reviewer #2: Yes

5. Review Comments to the Author

**Reviewer #1: ** 1. Co-occurrence analyses are usually used to quantify species loss and gain across communities, to investigate patterns of antibiotic cross-resistance, and to identify mechanistic similarities among diseases. Authors need to elaborate the use of co-occurrence analysis for attraction and repulsion between different inversions. This is particularly important as the focal theme of the study is to analyze the epistatic interaction between the inversions. Authors may add a note on the perceived advantage of using Jaccard-Tanimoto index over Sørensen-Dice, and Simpson methods.

2. For Table I, authors have used P-values as suggested by the analysis software, which means raw p-values have been utilized. This is not appropriate, kindly improve the presentation and add scientific P-values < 0.05." This is crucial as the co-occurrence analysis focuses only on the significant positive associations.

3. For the network analysis, some other libraries and approaches can be used to achieve the comprehended results of the Jaccard indexes and taken as a reference for building interaction. You can find alternatives and consider NetworkX in Python instead of igraph in R.

4. The techniques or the extensions authors have used vis a vis R language is not described in the method section. Authors need to emphasize and add a description regarding the usage of R or R Studio for the larger benefit of the readers.

5. Please indicate the type of inversions used for the analysis (paracentric / pericentric). Additionally, a note on the adaptive role or potential of the same need to be added in the discussion section.

6. Authors may consider the following references to incorporate in the revised manuscript:

(i) Chromosome inversions and linkage disequilibrium in Drosophila. Current Science 94:459-64.

(ii) Hundred years of research on inversion polymorphism in Drosophila. Current Science 117:761-775.

**Reviewer #2:**  Detailed comments are attached in a pdf file! Briefly:

The manuscript investigates epistatic interactions between genome inversions in Mimulus guttatus, focusing on whether these interactions exhibit non-additive effects on fitness. The authors adapt community ecology co-occurrence tests (e.g., chi-square, Jaccard/Tanimoto index J, and affinity score α) to identify deviations in inversion co-occurrence, suggesting epistasis. Network analysis is also employed to assess higher-order interactions.

Key findings:

1. A small subset (<10%) of inversion combinations deviate from additivity, though the direction of these interactions (positive/negative) is inconsistent.

2. Interactions are pairwise without evidence of higher-order effects.

Strengths:

1. A novel methodological framework.

2. Rigorous controls to eliminate confounding effects (e.g., physical linkage, Mendelian inheritance).

Major concerns:

1. [CRITICAL] The manuscript lacks engagement with established frameworks, like linkage disequilibrium (D).

2. Key metrics (e.g., J, α) and statistical methods (e.g., bootstrapping, sequential chi-square) require better explanation and justification.

Overall, the study provides a novel analytical approach to measure epistasis. Further clarifications and methodological comparisons are needed for robustness.

6. PLOS authors have the option to publish the peer review history of their article (what does this mean? ). If published, this will include your full peer review and any attached files.

**Do you want your identity to be public for this peer review?** For information about this choice, including consent withdrawal, please see our Privacy Policy .

Reviewer #1: No

Reviewer #2: **Yes: ** Harshavardhan Thyagarajan

---

## [Author Response · Author response to Decision Letter 1]

30 Jan 2025

Journal requirements

and

A/: We followed the guidelines, and the manuscript now should meet all the requirements.

A/: We fixed the problem. Right grant is now included: NSF grant MCB-1940785

A/: Fixed.

National Institute of General Medical Sciences (NIGMS/NIH) under award number P30GM145499.

A/: That financial statement was included by error. We are sorry.

4. Please note that your Data Availability Statement is currently missing the repository name. If your manuscript is accepted for publication, you will be asked to provide these details on a very short timeline. We therefore suggest that you provide this information now, though we will not hold up the peer review process if you are unable.

A/: Data Availability Statement added in the manuscript.

A/: Data is fully accessible without any restriction. It is part of the public repository where the code is stored.

6. Please amend either the abstract on the online submission form (via Edit Submission) or the abstract in the manuscript so that they are identical.

A/: Amendment done.

7. Please upload a new copy of Figure 3 as the detail is not clear. Please follow the link for more information: https://blogs.plos.org/plos/2019/06/looking-good-tips-for-creating-your-plos-figures-graphics/"
https://blogs.plos.org/plos/2019/06/looking-good-tips-for-creating-your-plos-figures-graphics/

A/: Figure was removed given that its information is covered with the linear models adjusted in the congruence between methods section. As suggested by one of the reviewers, this binarized analysis was omitted.

Editor Comments

Main points

1. Provide better explanation for the metrics, as they are novel to the PopGen community. Can be done with extended methods, or supplemental methods. In the intro, the network is presented as an alternative approach to the cooccurrence stats, but here it is used as a follow up analyses/visualization.

A/: This was an excellent suggestion. We included a supplemental methods file to cover the theoretical basis of the co-occurrence metrics employed. Also, the use of the network approach was clarified and improved.

2. Elaborate on strengths/weaknesses of these methods to detect coupling vs repulsion (suggest using the LD metric vocabulary).

A/: A brief discussion of the advantages and potential flaws of the indexes were included in the Materials and Methods section of the paper. Thank you so much for the suggestion.

3. Can you compare the performance of the metrics with established estimators, like linkage disequilibrium (D and D’). We are not ask for full scale benchmark analyses, but comparing the outputs of the different metrics can illustrate the nature of the signals they are best equipped to identify. Mention in discussion that benchmark analyses on simulated data with various properties (missingness, patterns of coupling, repulsion, recombination rates etc.) would be an ideal next step.

A/: This is a good point. We included a simulation-based contrast to illustrate the performance of these two metrics in relation to the LD framework. Nevertheless, it should be noted that LD metrics are best suited for haplotype-based analyses, where the data is analyzed at allelic level. In our case, we were analyzing the inversions at genotypic level. To perform a meaningful contrast, the co-occurrence metrics were applied at the allele level assuming HWE. It will be released as a complementary analysis to improve the integration with LD metrics.

As a summary, we can state that: our choice of co-occurrence metrics over LD metrics was motivated by both biological and statistical considerations. First, our analysis focuses on genotypic associations rather than haplotypic relationships, making co-occurrence metrics more directly applicable to our research question. Second, while our analysis shows that LD metrics can be stable (relative variation of D' = 0.238), they exhibit more complex relationships with allele frequencies (EDF ≈ 2.5) compared to co-occurrence metrics (EDF ≈ 1.0-1.3). This simpler frequency relationship makes co-occurrence metrics more interpretable for our genotype-level analysis.

4. Can you synchronize the terminology and labelling of the different methods? Like between Figure 1 and 2, and text for the simulation methods, it is a bit hard to follow which method is which. Perhaps it is just an issue of improving Figure 2 and its legend ? (don’t need to state “P-values“ in heading of all 9 panels).

A/: Yes, that is a good step forward. It is true that it is hard to follow the text without more clarification. The ‘p-value’ was removed from all the panels. Also, the organization was improved by adding a color-based system to identify the omnibus Chi-square test, the affinity-based results, and the Jaccard-based results. Same modifications were made to the supplementary figures.

5. Provide a well annotated R script (GitHub ideally) that explains the steps in the analyses, with well described (metadata) dataset. This will increase likelihood of the methods being picked up by other groups.

A/: The annotation of the main and auxiliary R scripts was improved. Thank you so much for the recommendation. They are all available in the GitHub repository released alongside the paper.

6. If the co-occurrence patterns depend strongly on Cross, then you are reporting higher order interactions. Cite literature on those (rather than organism specific papers), like https://www.sciencedirect.com/science/article/pii/S0168952513000218?via%3Dihub

A/: The paper you recommended was extremely useful for strengthening the discussion of our results (cite added) related to the context dependent nature of the interactome. Thank you so much.

Intermediate points

1. Think it may be helpful to generate a figure for the data on 336-242, to demonstrate how the numbers look for the 3x3 tables. This may provide a better link to the graphics following! Your call!

A/: Yes, it will be a great addition to a representation of the data. It is included now in Fig 1.

2. Check wording and grammar of the entire manuscript, see our minor points!

A/: Checking of grammar and wording done.

Minor points

1. Line 45: Don’t agree with the last part of sentence “In genetics, non-random relationships between loci, described as linkage disequilibria, are measures of co-occurrence”. The non random relationships are not measures!

A/: We agree. Wording corrected.

2. Line 49: Can you reword “Attraction or repulsion of species in their location.”

A/: We reworded the sentence.

3. Line 58: Metrics?? “The co-occurrence techniques.”

A/: Most definitely ‘metrics’ is the word to use here. Thank you.

4. Line 64: Drop “of” “In our study species of Mimulus guttatus.”

A/: We did so.

5. Line 77: Provide references ”Network analysis is an area of active development in biology.”

A/: Two references were added to support the statement.

6. Line 79: Singular or plural idea(s)? “The underlaying basic ideas.”

A/: It should be in singular. It was fixed.

7. Line 104: Which 10 genomes? “The ten genomes were derived.”

A/: Statement clarified. Those genomes were employed to detect the inversions employed in this paper.

8. Line 130: “if genotype frequencies” these are all inversion genotypes?

A/: Statement clarified. Now it reads: “ To test for segregation distortion (SD) at individual inversion loci (Fig 1A), we performed a goodness-of-fit test to determine if inversion genotype frequencies (standard homozygous, heterozygous, and inverted homozygous) followed expected Mendelian proportions for an F2 population (0.25/0.50/0.25).”

9. Line 205: How many datapoints were in this comparison dataset? “also detect signals derived from linkage.”

A/: The number of data points in this case is simply the number of valid tests we performed in the 2x2 framework. A note in the figure legend was added to clarify this point. In relation with the linkage (in the sense of physical linkage), if the results for inversions in the same chromosome are explored, it can be seen how these tend to co-occur more frequently than by chance. Those inversions have elevated and positive scores for affinity (value 10 or very close to 10, being 10 the capped maximum in the log ratios test) and centered Jaccard / Tanimoto. We do not include those results in the paper because we omit them, because physical linkage alone can explain these interactions (we only care for interchromosomal interactions)

10. Line 211: Used rather than adjusted? “we adjusted robust linear regression instead.”

A/: Wording fixed following your recommendation.

11. Line 247-251: Reword this section, introduce the permutation approach earlier in the description.

A/: Suggestion implemented following your suggestion. Permutation approach moved before some other support statements related to the decision we made.

12. Line 258: “in six other crosses where it displays no distortion.” Or state directly, “where both types of homozygotes are seen”?

A/: Statement re-worded.

13. Line 274: Table 1. Can you annotate the p values in different way? X21 followed by column with its p values, and then X22? And a column that has all the same values is useless, i.e. “Selection”

A/: Modifications made following your suggestions.

14. Line 276: Maybe reword title to “Testing of methods via simulation(s??)”?

A/: Recommendation of title adopted. Thank you so much.

15. Line 286: Maybe say “counts” were “deflated”, not cell?

A/: Fixed.

16. Line 309: “Of the valid contrasts…” sentence is convoluted. Can this be subdivided into 2 or more sentences or reworded?

A/: Fixed. Now it reads: “Among contrasts that distinguish between homozygous and heterozygous inversion states, we detected 355 significant associations (~9% of valid combinations) using at least one of three analytical approaches: Jaccard similarity, affinity coefficients, or post-hoc tests (based on uncorrected p-values).”

17. Line 318: Check reference to supplement II.

A/: Checked!

18. Line 320: Any cases of the opposite? “but the” instead of “and the”?

A/: Suggestion implemented. In relation with the opposite scenario, it can be seen in the Fig 4, for those edges that are significant for the Chi_squared test but not for the Jaccard index.

19. Line 366: Was there good or poor congruence? Consider changing the section heading “Congruence between methods” or first sentence “The congruence between the 2x2 table-based methods is depicted in Fig. 3 and Fig. 3S.” to better explain the main result of this section.

A/: There was a good congruence between methods (now added to the results). The section was re-worded following your suggestions. Thank you so much. Notice that congruence now is assessed using the linear regression results, following suggestions of one of the reviewers (binarized analysis are not necessary to show the point).

20. Line 371: “robust”, is this word needed? Anything specifically robust about the regression?

A/: Yes, in this context, it refers to a regression analysis designed to be less sensitive to outliers and violations of statistical assumptions. It was necessary to implement it given that the residuals of the ordinary regression implemented first were not normally distributed.

21. Line 367 and Figure 3: It is unclear what is shown in the figure (legend missing details, what is above and below diagonal. Labels on figure to small to read, question if inset color panel is needed for all 9? Make start of Figure 3 legend more transparent “good correspondence was between…”

A/: Figure is no longer part of the results. It message can be conveyed directly with the regression results in Supporting Information.

22. Line 384: Isn’t this more of a visualization of the results from the previous section?? Heading “Visualization of co-occurrence as inversion networks” or some version of this?

A/: True. Formal network analysis included! Following reviewer suggestions, now we analyzed the motifs in the networks to search for significant patterns (third and fourth order motif included).

23. Line 390: Question of wording ”there were significant patterns”, maybe “signal”?

A/: Phrase re-worded.

24. Line 391: “Strongest signals were in lines 155”

A/: Suggestion implemented.

25. Line 439: “individuals” rather than “organisms”

A/: Change made following suggestion.

26. Line 442: Wording, “Also, inversions can be central elements in our understanding about”, maybe drop the “our understanding”?

A/: “our understanding” was dropped.

27. Line 453: “interactome” is not the right word here, find another. Also fix in other places in MS.

A/: We agree. Interactome has been adopted in a molecular sense for protein-protein interactions and so forth. We replaced the word all along the manuscript.

28. Line 456: “induced” another verb may be better!

A/: Change made following your suggestion.

29. Line 470: Reword “although often not across all crosses” was it not so that this was mainly in few crosses?

A/: Change made following your suggestions.

30. Line 479: Check sentence structure and placement of reference.

A/: Fixed.

31. Line 482: Strange wording “…where the higher count of plants was found in the homozygous”. Can you rephrase. “more plants were homozygous for inversion 49 …”?

A/: Fixed following your suggestion.

32. Line 485: “effective population size”?

A/: Yes. Fixed following your suggestion.

33. Line 491: Wording? “Applied to in this study”

A/: Wording fixed.

34. Line 507: Add citations to papers about higher order epistasis, see above.

A/: The citations were added. Thank you very much.

35. Line 511: “We investigated one such example in detail” rather than “Figure 5 provides a particular example of variability of interactions.” Or some other more descriptive version of this.

A/: Wording fixed.

36. Line 518: Reword, in particular clunky text at “In the case of the inversion 32 in homozygous state and the inversion 40 in heterozygous state, it can be seen a significant effect for the heterogeneity”

A/: Statements re-worded.

37. Line 528: “strong” not “string”?

A/: Fixed.

38. Line 530: Verb missing…”are?”

A/: Fixed.

39. Line 533: Maybe we just need more power, or that the decencies of SDV and inversion interactions are d

---

## [Decision Letter · Decision Letter 1]

14 Feb 2025

PONE-D-24-52521R1Are you with me? Co-occurrence tests from community ecology can identify positive and negative epistasis between inversions in Mimulus guttatusPLOS ONE

Dear Dr. Madrigal-Roca,

Thank you for submitting your manuscript to PLOS ONE. After careful consideration, we feel that it has merit but does not fully meet PLOS ONE’s publication criteria as it currently stands. Therefore, we invite you to submit a revised version of the manuscript that addresses the points raised during the review process.

The manuscript is now greatly improved and nearing completion. There are a few editorial issues outstanding, the main one concerns the supplemental methods and LD comparison. Rev 2 did not have access to this, which explains some of the comments, but there are still some issues.

These are the most compelling. The LD addition is an improvement, “We included a simulation-based contrast to illustrate the performance of these two metrics in relation to the LD framework” but I think this should be described in the manuscript (see rev 2). The approach is described in the supplemental section, but I suggest moving it to the main MM. (“Comparison between LD and co-occurrence metrics in an allele-based analysis… fixed association strength of 0.5.”).

And then include a brief summary, like in the RespRev section, to describe the main findings in the manuscript proper. Also, the results of these analyses (points 1-4) in supplemental section, do not include figures or data. Suggest you add that (like 4. “Heatmap plot showing…”). Finally, please define EDF.

Second main thing is language, it needs improvement. We strongly encourage the senior author to read and copy edit the MS thoroughly in this edition. At PLOS it is not in the reviewers or editors role to improve the language.

We look forward to receiving your revised manuscript.

Kind regards,

Arnar Palsson, Ph.D.

Academic Editor

PLOS ONE

Journal Requirements:

**Additional Editor Comments:**

The manuscript is now greatly improved and nearing completion. There are a few editorial issues outstanding, the main one concerns the supplemental methods and LD comparison. Rev 2 did not have access to this, which explains some of the comments, but there are still some issues.

These are the most compelling. The LD addition is an improvement, “We included a simulation-based contrast to illustrate the performance of these two metrics in relation to the LD framework” but I think this should be described inthe manuscript (see rev 2). The approach is described in the supplemental section, but I suggest moving it to the main MM. (“Comparison between LD and co-occurrence metrics in an allele-based analysis… fixed association strength of 0.5.”).

And then include a brief summary, like in the RespRev section, to describe the main findings in the manuscript proper. Also, the results of these analyses (points 1-4) in supplemental section, do not include figures or data. Suggest you add that (like 4. “Heatmap plot showing…”). Finally, please define EDF.

Second main thing is language, it needs improvement. We strongly encourage the senior author to read and copy edit the MS thoroughly in this edition. At PLOS it is not in the reviewers or editors role to improve the language.

Intermediate

The annotations of columns, terms and p values still varies from table to table, and in tables and the footings. Please carefully check all tables, tidy this up and standardize.

Can you recode large and small numbers, e.g. from 3.29E-30 to more classical annotation 3.29 x 10-30

In text, tables and figures.

Minor points:

Line 17. Maybe insert “unlinked” into the sentence?

Line 56. “Co-occurrence in geneticS…”

Line 68 should this be singular or plural, one method or many?

Line 78 on not of “of 4.5mb on chromosome 6”

Line 83 sentence should end with a citation!

Line 88 pleasre rephrase “Network analysis is an area of active development in biology [20,21].”

Line 203 which metrics ? “between two widely used LD metrics”

Line 206. Figure 2. Is the coupling case – was it deliberate to have one individual of the “fourth type”? In the figure as it “stands”, column 5, line 2.

Legend of table 1, same X2 annotation for both, no explanation of which is which.

Is table 2 justified? Is another representation of the data more helping. This appears a bit like a supplemental table, or is it a list of interesting things?

Figure 3

The scale of X is confusing, can you improve it?

Line 377. “While most inversion pairs show no evidence of interaction, there are interesting exceptions.” Is there maybe more important summary statement at start. Are there different types of patterns to the associations??

Line 387

“The most frequent deviations from independence were observed in heterozygous-homozygous combinations (~58%),” Is this an excess over the expecations. Which fraction of the combinations tested are het-homo? Consider same descriptions for other combinations.

Line 399

Fig missing “as depicted in 3S”?

Line 411

“S trongest signals were more abundant in lines 155, 502, 541, 62, 664, and 909, while in the rest they were scarcer.” Does this rhyme with the JT analyses?

Check title supplemental data file “suplemental”

Also, from supplemental methods file, please cross reference to the manuscript for particulars (methods, tables, figures).

Reviewers' comments:

Reviewer's Responses to Questions

**Comments to the Author**

1. If the authors have adequately addressed your comments raised in a previous round of review and you feel that this manuscript is now acceptable for publication, you may indicate that here to bypass the “Comments to the Author” section, enter your conflict of interest statement in the “Confidential to Editor” section, and submit your "Accept" recommendation.

Reviewer #1: All comments have been addressed

Reviewer #2: (No Response)

2. Is the manuscript technically sound, and do the data support the conclusions?

Reviewer #1: (No Response)

Reviewer #2: Yes

3. Has the statistical analysis been performed appropriately and rigorously? 

Reviewer #1: (No Response)

Reviewer #2: Yes

4. Have the authors made all data underlying the findings in their manuscript fully available?

Reviewer #1: (No Response)

Reviewer #2: Yes

5. Is the manuscript presented in an intelligible fashion and written in standard English?

Reviewer #1: (No Response)

Reviewer #2: Yes

6. Review Comments to the Author

Reviewer #1: (No Response)

Reviewer #2: I am satisfied that meaningful improvements have been made through revisions in line with the AE’s guidance. While their engagement with LD remains somewhat underdeveloped / could be better articulated, it is an improvement over the previous version. A more direct comparison using LD as a metric—supported by statistical phasing of haplotypes—would have resulted in much stronger understanding. Although not an obstacle to scientific understanding, the language in the manuscript still requires substantial proofreading. I recommend accepting, given some minor revisions, for eg. acknowledging certain caveats. In addition, the extended methods section of the supplement appear to be missing, along with the details of how they carried out the simulation based comparison, and should be included prior to acceptance.

Detailed comments in attached document.

7. PLOS authors have the option to publish the peer review history of their article (what does this mean? ). If published, this will include your full peer review and any attached files.

**Do you want your identity to be public for this peer review?** For information about this choice, including consent withdrawal, please see our Privacy Policy .

Reviewer #1: No

Reviewer #2: No

---

## [Author Response · Author response to Decision Letter 2]

28 Feb 2025

Additional Editor Comments:

The manuscript is now greatly improved and nearing completion. There are a few editorial issues outstanding, the main one concerns the supplemental methods and LD comparison. Rev 2 did not have access to this, which explains some of the comments, but there are still some issues.

A/: All extended methods and supplementary results were carefully inspected and included in submission. We ask to keep them as supplementary elements, given the number of results we already have.

These are the most compelling. The LD addition is an improvement, “We included a simulation-based contrast to illustrate the performance of these two metrics in relation to the LD framework” but I think this should be described inthe manuscript (see rev 2). The approach is described in the supplemental section, but I suggest moving it to the main MM. (“Comparison between LD and co-occurrence metrics in an allele-based analysis… fixed association strength of 0.5.”).

A/: We would prefer to maintain the simulation exercise for contrasting LD measurements and co-occurrence metrics as supplementary materials. These demonstration is based on hypothetical inversions, analyzed from the allele frequency perspective, to make them comparable with LD metrics. Additionally, the use of co-occurrence metrics in the manuscript is directed to genotypic frequencies.

And then include a brief summary, like in the RespRev section, to describe the main findings in the manuscript proper. Also, the results of these analyses (points 1-4) in supplemental section, do not include figures or data. Suggest you add that (like 4. “Heatmap plot showing…”). Finally, please define EDF.

A/: The supplementary materials containing these results were improved in organization. Also, EDF (effective degrees of freedom) was defined in the Extended Methods.

Second main thing is language, it needs improvement. We strongly encourage the senior author to read and copy edit the MS thoroughly in this edition. At PLOS it is not in the reviewers or editors role to improve the language.

A/: Senior and junior scientists went over the language one more time to improve it. Thank you so much for the suggestions.

Intermediate

The annotations of columns, terms and p values still varies from table to table, and in tables and the footings. Please carefully check all tables, tidy this up and standardize.

A/: The annotation of the columns was standardized. Thank you so much for the suggestion.

Can you recode large and small numbers, e.g. from 3.29E-30 to more classical annotation 3.29 x 10-30

In text, tables and figures.

A/: Scientific notation changed to classical representation.

Minor points:

Line 17. Maybe insert “unlinked” into the sentence?

A/: Done.

Line 56. “Co-occurrence in geneticS…”

A/: Corrected.

Line 68 should this be singular or plural, one method or many?

A/: Changed to “These metrics …”

Line 78 on not of “of 4.5mb on chromosome 6”

A/: Corrected.

Line 83 sentence should end with a citation!

A/: Citation included.

Line 88 pleasre rephrase “Network analysis is an area of active development in biology [20,21].”

Line 203 which metrics ? “between two widely used LD metrics”

A/: Changes were implemented according to your suggestions.

Line 206. Figure 2. Is the coupling case – was it deliberate to have one individual of the “fourth type”? In the figure as it “stands”, column 5, line 2.

A/: We do not understand the question.

Legend of table 1, same X2 annotation for both, no explanation of which is which.

A/: The issue was corrected.

Is table 2 justified? Is another representation of the data more helping. This appears a bit like a supplemental table, or is it a list of interesting things?

A/: Yes, those inversion genotype pairs are one of the main results of the paper. A set of potential combination with coupling and repulsion effects worth further investigation.

Figure 3

The scale of X is confusing, can you improve it?

A/: The scale was changed by using classical scientific notation.

Line 377. “While most inversion pairs show no evidence of interaction, there are interesting exceptions.” Is there maybe more important summary statement at start. Are there different types of patterns to the associations??

A/: The text was rearranged to present the summary with the different patterns of association between dosages levels first.

Line 387

“The most frequent deviations from independence were observed in heterozygous-homozygous combinations (~58%),” Is this an excess over the expectations. Which fraction of the combinations tested are het-homo? Consider same descriptions for other combinations.

A/: We did not test for an excess of combinations in this case. We did not have a reason to do it. It could be worth it with further data, to see if there is a consistent bias. In terms of proportions, given that for any given pair of inversion, you have two dosage levels in consideration (heterozygous and homozygous), you would have two contrasts heterozygous-homozygous, one heterozygous-heterozygous, and one homozygous-homozygous (i. e. Inv_1_1 vs Inv_2_2; Inv_1_2 vs Inv_2_1; Inv_1_1 vs Inv_2_1; Inv_1_2 vs Inv_2_2). So, ½ of the contrast are heterozygous-homozygous. We did not report again the proportion relative to all the contrast performed because we presented the information related to how many relevant tests from the total we were interested in, and now we say, from those potential contrasts, which ones reflect a deviation from independence in terms of dosage combinations.

Line 399

Fig missing “as depicted in 3S”?

A/: Corrected.

Line 411

“S trongest signals were more abundant in lines 155, 502, 541, 62, 664, and 909, while in the rest they were scarcer.” Does this rhyme with the JT analyses?

A/: Yes, in the context of the paragraph, that opens the presentation of the networks per cross.

Check title supplemental data file “suplemental”

Also, from supplemental methods file, please cross reference to the manuscript for particulars (methods, tables, figures).

A/: Everything was reviewed and cross-referenced.

Reviewer 2

Major issue #1: Engagement with LD, comparative analysis.

As far as I can see, neither simulation methods/parameters nor the calculations used to make

subsequent measurements are described in the supplement. I am unable to follow what has been

reported.

A/: The supplementary materials and extended methods were better labeled for identification in the current submission.

Major issue #2: Insufficient explanations

The authors add a description in the introduction2, and a toy example as promised through fig.2.

However, the status of the 'Extended Methods' section is unclear. I am not able to locate such a

section in the “supporting information” file. Perhaps this was intended as a supplementary

section but omitted accidentally.

As noted above, the extended methods were not available for review.

A/: The supplementary materials and extended methods were better labeled for identification in the current submission.

---

## [Editor Report · Decision Letter 2]

4 Mar 2025

Are you with me? Co-occurrence tests from community ecology can identify positive and negative epistasis between inversions in Mimulus guttatus

PONE-D-24-52521R2

Dear Dr. Madrigal-Roca,

We’re pleased to inform you that your manuscript has been judged scientifically suitable for publication and will be formally accepted for publication once it meets all outstanding technical requirements.

Kind regards,

Arnar Palsson, Ph.D.

Academic Editor

PLOS ONE
---

## [Editor Report · Acceptance letter]

PONE-D-24-52521R2

PLOS ONE

Dear Dr. Madrigal-Roca,

I'm pleased to inform you that your manuscript has been deemed suitable for publication in PLOS ONE. Congratulations! Your manuscript is now being handed over to our production team.

Kind regards,

on behalf of

Dr. Arnar Palsson

Academic Editor

PLOS ONE